# ParallelComp: Parallel Long-Context Compressor for Length Extrapolation

Jing Xiong [*1]  Jianghan Shen [*2]  Chuanyang Zheng [3]  Zhongwei Wan [4]  Chenyang Zhao [5]  Chiwun Yang [6]
Fanghua Ye [7]  Hongxia Yang [8]  Lingpeng Kong [1]  Ngai Wong [1]

## Abstract

Extrapolating ultra-long contexts (text length > 128K) remains a major challenge for large language models (LLMs), as most training-free extrapolation methods are not only severely limited by memory bottlenecks, but also suffer from the attention sink, which restricts their scalability and effectiveness in practice. In this work, we propose PARALLELCOMP, a parallel long-context compression method that effectively overcomes the memory bottleneck, enabling 8B-parameter LLMs to extrapolate from 8K to 128K tokens on a single A100 80GB GPU in a training-free setting. PARALLELCOMP splits the input into chunks, dynamically evicting redundant chunks and irrelevant tokens, supported by a parallel KV cache eviction mechanism. Importantly, we present a systematic theoretical and empirical analysis of attention biases in parallel attention—including the attention sink, recency bias, and middle bias—and reveal that these biases exhibit distinctive patterns under ultra-long context settings. We further design a KV cache eviction technique to mitigate this phenomenon. Experimental results show that PARALLELCOMP enables an 8B model (trained on 8K context) to achieve 91.17% of GPT-4's performance under ultra-long contexts, outperforming closed-source models such as Claude-2 and Kimi-Chat. We achieve a 1.76x improvement in chunk throughput, thereby achieving a 23.50x acceleration in the prefill stage with negligible performance loss and pave the way for scalable and robust ultra-long contexts extrapolation in LLMs. We release the code at https://github.com/menik1126/ParallelComp.

*Equal contribution [1]The University of Hong Kong, [2]Nanjing University, [3]The Chinese University of Hong Kong, [4]The Ohio State University, [5]The University of California, Los Angeles, [6]Sun Yat-Sen University, [7]Tencent, [8]Hong Kong Polytechnic University. Correspondence to: Jing Xiong < junexiong@connect.hku.hk >.

*Proceedings of the 42nd International Conference on Machine Learning*, Vancouver, Canada. PMLR 267, 2025. Copyright 2025 by the author(s).

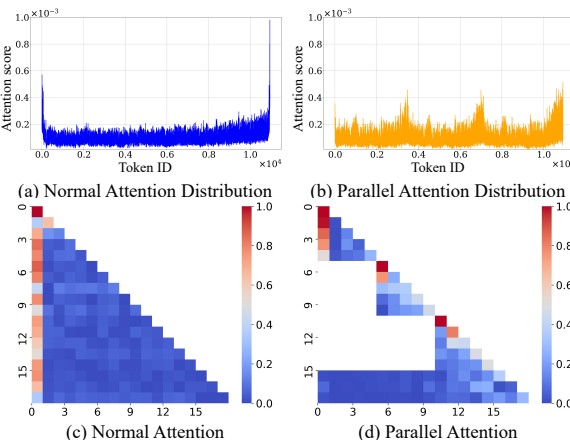

Figure 1: **Upper:** Distribution of two types of attention. **Lower:** Heatmaps of the two types of attention. The attention head shown here represents the distribution change between normal attention and parallel attention for the 21st head in layer 1, using the Llama-2-7b-chat-hf model.

## 1. Introduction

Extrapolating long-contexts is a core capability of large language models (LLMs) (Achiam et al., 2023; Touvron et al., 2023; Dubey et al., 2024). Achieving this requires effective mechanisms for modeling positional relationships over extended sequences. Rotary position embedding (RoPE) (Su, 2023) are commonly employed in LLMs due to their ability to efficiently encode relative positional information. However, extrapolating to lengths beyond the training range during inference remains a significant challenge. While retraining or fine-tuning the entire model is one possible solution, it is not always feasible, particularly in resource-constrained environments (Press et al., 2022; Chi et al., 2022; Peng et al., 2023; Fu et al., 2024). This limitation prevents RoPE-based models from generalizing to longer sequences without additional training, making them less adaptable to scenarios requiring inference on inputs of unforeseen lengths.

To address this, many researchers explore training-free approaches for context length extrapolation. One prominent direction involves incorporating NTK-based methods (LocalLLaMA, 2023b;a; bloc97, 2023), which distinguish dimensions of different frequencies by their wavelengths and

apply tailored interpolation strategies to address the extrapolation challenge of position encoding. Another promising strategy leverages text chunking techniques (An et al., 2024; Xiao et al., 2024; Ratner et al., 2022; Zhu et al., 2024), which typically reuse position encodings across chunks to enable chunk-wise length extrapolation without retraining, thereby improving the model's ability to generalize to longer contexts. However, both approaches suffer from the attention sink phenomenon (Liu et al., 2024; Xiao et al., 2023; Han et al., 2024; Gu et al., 2024), where high attention scores tend to be assigned to the first few tokens or the last few tokens in the input sequence. Figure 1 illustrates this phenomenon and compares the attention distributions between standard and parallel attention mechanisms. In the upper part of Figure 1, we observe the distribution of attention scores across tokens. In the standard attention distribution (left), corresponding to the heatmaps (c) below the image, attention is disproportionately focused on the first and last few tokens. In contrast, the parallel attention distribution (right), shown in the heatmaps (d) below the image, attempts to distribute attention more evenly. However, it still shows noticeable concentrations in certain regions and exhibits a multi-peak distribution, indicating that the nature of the attention sink differs fundamentally between the two mechanisms. The phenomenon within the length extrapolation mechanism in the parallel attention mechanism remains unexplored. This paper focuses on addressing the memory limitations encountered during length extrapolation and provides a detailed analysis of the unique attention sink that emerges in parallel attention. Specifically, we extrapolate the length by chunking the input and reusing position encodings, applying parallel compression of the KV cache of the chunks and tokens to resolve memory bounds, and exploring the unique phenomenon of the attention sink in parallel attention while attempting to adopt a token eviction strategy to mitigate this bias. To promote the understanding of the effects of special attention sink on parallel attention, we propose the following questions in this paper: **Q1:** *What types of attention patterns can be summarized?* **Q2:** *Is there any difference between the attention bias in parallel attention and the attention bias in classical attention?* **Q3:** *Can the calibration strategy alleviate attention bias?* **Q4:** *Does our parallel compression strategy effectively support length extrapolation?*

To address the above questions, we propose PARALLEL-COMP, a parallel long-context compression method to extrapolate length. While maintaining high throughput, we extrapolate the length from 8K to 128K on a single GPU, with almost no performance loss. Overall, our contributions are as follows:

- We propose PARALLELCOMP, a novel training-free method that enables efficient length extrapolation for

LLMs — scaling from 8K to up to 128K tokens on a single A100 80GB GPU — by addressing attention biases through an effective attention calibration strategy.

- To overcome limitations in parallel attention, we introduce a chunk eviction mechanism and parallel KV cache eviction, allowing processing of contexts beyond 128K tokens and achieving a 1.76x throughput improvement and a 23.50x speedup in the prefilling stage, with negligible performance loss.

- Experiments demonstrate that PARALLELCOMP achieves **91.17% of GPT-4's performance** on ultra-long context tasks using an 8B model trained on only 8K-length context, outperforming strong proprietary models such as Claude-2 and Kimi-Chat.

## 2. Related Work

**Position encoding.** Existing absolute position encoding (APE) (Vaswani, 2017; Devlin, 2018) incorporates either fixed or learnable position encodings into input representations through vector addition. However, APE faces challenges when dealing with long-contexts. To overcome these limitations, relative position encoding (RPE) methods—such as rotary and additive position encodings (Su, 2023; Su et al., 2024; Press et al., 2021)—are developed to offer improved generalization in longer contexts. In addition, some data-dependent position encoding methods (Golovneva et al., 2024; Zheng et al., 2024) are gaining widespread attention. These position encodings are generated based on the input data.

Some works (Press et al., 2022; Chi et al., 2022; Li et al., 2023a) also focus on designing better position encodings to enhance the pre-trained model's capability for length extrapolation. Another line of works (Peng et al., 2023; bloc97, 2023; Chen et al., 2023a) focus on enhancing the LLM's length extrapolation capability by fine-tuning.

Furthermore, there are two categories of training-free extrapolation methods. The first category, such as LocalLLaMA (2023b); Chen et al. (2023b); LocalLLaMA (2023a); Chen et al. (2024), directly modifies position encodings to enable extrapolation or interpolation, aiming to enhance the model's length extrapolation capability. The second category (An et al., 2024; Xiao et al., 2024; Ratner et al., 2022; Zhu et al., 2024) achieves extrapolation solely by reusing position encodings. In this work, we focus primarily on the training-free setting.

**Attention sink.** A series of studies (Gu et al., 2024; Xiao et al., 2023; Liu et al., 2024) reveal the phenomenon of attention sink, where certain tokens in the sequence (referred to as sink tokens) consistently receive abnormally high attention scores.

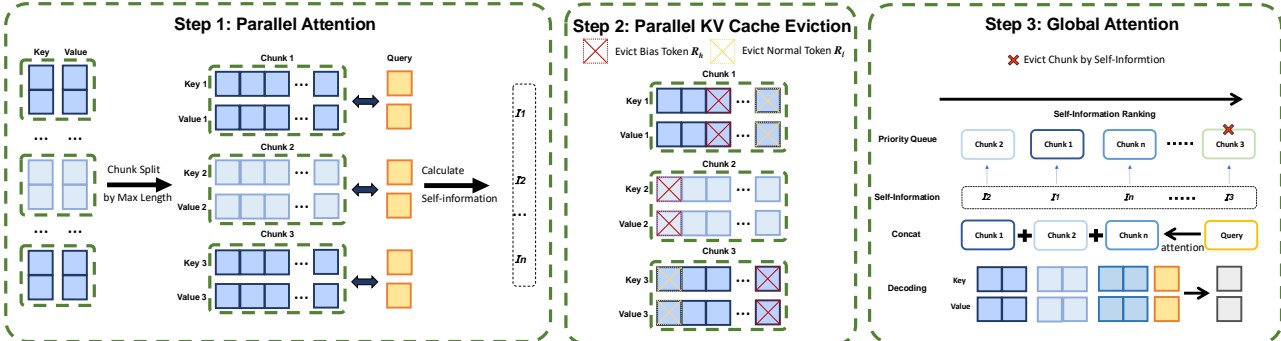

Figure 2: Overview of PARALLELCOMP. **Parallel Attention** – The input sequence is split into multiple chunks based on the model's maximum context length. Each chunk undergoes local attention computation independently, and the self-information score of the query is calculated. **Parallel KV Cache Eviction** – Based on the self-information score, low-score tokens (marked in yellow, $R_L^h$) and attention bias tokens (marked in red, $R_H^h$) are selectively evicted to optimize memory usage and attention bias. **Global Attention** – The remaining KV caches are ranked by self-information, and less relevant chunks are discarded. The selected chunks are then concatenated, and a global attention operation is applied to ensure comprehensive information aggregation before the final autoregressive decoding stage.

When the sequence length increases, the attention scores in the middle of the sequence are significantly lower compared to the beginning and the end. Some work (Xiao et al., 2023; Zhang et al., 2023; Wan et al., 2024; Xiong et al., 2024) find that the tokens of attention sink can effectively aggregate information and use it to efficiently compress the KV cache. But in ultra-long contexts, this may prevent the model from focusing on the correct parts of long sequences. Xiao et al. (2023) attributes this phenomenon to the softmax function, which forces the query to assign attention scores to all preceding tokens, even when the preceding tokens lack essential information, resulting in high scores being assigned to initial tokens. Gu et al. (2024) proposes a simple solution by replacing the softmax attention with alternative attention mechanisms (e.g., unnormalized sigmoid attention) to alleviate this dependency. Chen et al. (2024) alleviates this phenomenon by simply removing certain low-frequency components from RoPE. Yu et al. (2024) deals with this issue by calibrating the attention distribution. In our work, we focus primarily on three phenomena of attention bias within parallel attention: the attention sink at the beginning of the input, the attention sink at the end of the input (i.e. recency bias Peysakhovich & Lerer 2023), and the scattered attention in the middle of the input (Yu et al., 2024). These biases provide insights into how LLMs utilize parallel contextual information.

## 3. Method

In this section, we introduce PARALLELCOMP, our proposed approach for achieving efficient long-context compression during extrapolation. We then discuss its unique

bias phenomenon. Figure 2 offers a high-level overview of PARALLELCOMP.

### 3.1. ParallelComp

**Parallel attention.** Inspired by previous studies (An et al., 2024), we divide the text into chunks based on the model's maximum context size and concatenate them with the input query for parallel encoding. This step is typically carried out using local attention. For a given input sequence $X \in \mathbb{R}^{N \times d}$, the sequence is split into $C = \lceil N/w \rceil$ chunks, each containing at most $w$ tokens (the maximum context length per chunk). Let $f_Q(\cdot)$, $f_K(\cdot)$, and $f_V(\cdot)$ denote the linear transformation functions for the query, key, and value projections, respectively. Attention computation is then performed within each chunk:

$$A_{\mathfrak{f}}^c = \text{Softmax}\left(\frac{f_Q(X^c) \cdot f_K(X^c)^T}{\sqrt{d}}\right), \quad (1)$$

where $X^c \in \mathbb{R}^{w \times d}$ represents the $c$-th chunk of the input sequence and $A_{\mathfrak{f}}^c \in \mathbb{R}^{w \times w}$ is the corresponding attention matrix. The feature update is performed for each chunk:

$$F^c = A_{\mathfrak{f}}^c \cdot f_V(X^c), \quad (2)$$

where $F^c \in \mathbb{R}^{w \times d}$ is the updated feature for the $c$-th chunk.

Below, we discuss how to design chunk eviction strategies and parallel KV cache eviction strategies to maintain high throughput while minimizing redundant computations.

**Chunk eviction.** To ensure that the computation of parallel attention can be performed on a single 80GB A100

GPU, we design a chunk eviction strategy to control memory overhead as shown in Figure 2 step 3. Inspired by Li et al. (2023b); Jiang et al. (2023), we introduce a chunk eviction mechanism that leverages the self-information of the query tokens $X^q$ to further enhance parallel processing efficiency. This mechanism employs an online priority queue to manage memory, retaining only the most relevant chunks with the lowest perplexity, thereby improving language modeling. For a given chunk $c$, the self-information score for the query tokens $X^q$ is calculated as follows:

$$I_c = -\log P(X^q \mid X^c), \tag{3}$$

where $X^c$ represents the context of chunk $c$ and $X^q$ corresponds to the chunk of the query. Chunks with lower self-information scores are considered more relevant and are retained. The set of indices $c$ corresponding to the selected chunks is denoted by:

$$S = \{c \mid c \le \epsilon\}, \tag{4}$$

where $\epsilon$ is a threshold that determines whether a chunk will be selected or not. The selected chunks are sorted based on $I_c$, and only a fixed number of top-ranked chunks are retained. These selected chunks are stored in a fixed-size priority queue to ensure that the prefilling stage remains within the memory limit.

**Parallel KV cache eviction.** To further improve chunk throughput, we propose a KV cache eviction strategy, as illustrated in Step 2 of Figure 2. We leverage Flash Attention (Dao, 2023) for efficient attention computation. Prior to performing local attention, we use the cumulative scores of $X^q$ to quickly identify tokens with relatively low attention importance and evict them. Specifically, the local attention score $A_l^c$ between the $i$-th token of the query $X_i^q$ and the $j$-th token of the chunk $X_j^c$ is calculated as:

$$A_{l(i,j)}^c = f(X_i^q, X_j^c), \tag{5}$$

where $f$ represents the matrix multiplication used to compute the attention score between $X_i^q$ and $X_j^c$. Then, the cumulative attention score for the $j$-th token in the $c$-th chunk is computed by summing the local attention scores over all tokens in the query chunk:

$$S_{c,j} = \sum_{i=1}^{w_q} A_{l(i,j)}^c, \quad j = 1, 2, ..., w, \tag{6}$$

where $S_{c,j} \in \mathbb{R}^w$ denotes the cumulative attention score for the $j$-th token in the $c$-th chunk, and $w_q$ is the length of the query chunk $X^q$. The cumulative attention score aggregates the attention distributions from each token in the query to each token in the chunk, thereby measuring the relevance of

each token in the chunk to the query $X^q$. Tokens with low cumulative attention scores within the chunk are evicted, and the retained tokens are used to form the compressed KV cache:

$$K_r^h = K_x^h[R_L^h], \quad V_r^h = V_x^h[R_L^h], \tag{7}$$

where $K_x^h$ and $V_x^h$ represent the KV cache of the $h$-th head of the input chunk, and $R_L^h$ denotes the set of indices corresponding to the evicted tokens in the $h$-th head with low attention scores. The notation $[\cdot]$ indicates indexing into $K_x^h$ and $V_x^h$ to evict only the tokens corresponding to the indices in $R_L^h$. $K_r^h$ and $V_r^h$ denote the retained KV cache. The compression strategy of the KV cache typically helps reduce memory overhead while increasing chunk throughput, but it often exacerbates attention bias. Next, we introduce a simple and effective strategy to calibrate attention distribution.

**Attention calibration.** To mitigate the attention bias exacerbated by *parallel KV cache eviction*, we propose an alternative token eviction strategy based on Eq. 6. Specifically, we evict tokens with excessively high attention scores. Let $R_H^h$ represent the tokens with attention scores of $h$-th head exceeding a manually-set threshold $\lambda$. Thus, we have:

$$K_{r'}^h = K_x^h[R_H^h], \quad V_{r'}^h = V_x^h[R_H^h]. \tag{8}$$

Evicting tokens with exceptionally high scores guarantees that the attention mechanism can produce calibrated attention distributions. We will thoroughly investigate the impact of this calibration method on the attention bias in Section 4.

**Global attention.** After obtaining the attention outputs for each chunk, we concatenate the KV caches from all chunks into a unified representation. Specifically, the concatenated KV cache is given by:

$$\begin{aligned} K &= \left[K^{X^1}, K^{X^2}, \ldots, K^{X^C}, K^{X^q}\right], \\ V &= \left[V^{X^1}, V^{X^2}, \ldots, V^{X^C}, V^{X^q}\right], \end{aligned} \tag{9}$$

where $K^{X^c}$ and $V^{X^c}$ are the KV caches of the $c$-th chunk.

Next, we perform a global attention operation. This global attention enables the model to aggregate information across all chunks, ensuring that global dependencies are captured. The global attention computation for $X^q$ is given by:

$$A_{\mathfrak{g}} = \text{Softmax}\left(\frac{f_Q(X^q) \cdot K^T}{\sqrt{d}}\right), \tag{10}$$

where $A_{\mathfrak{g}} \in \mathbb{R}^{w_q \times (C \cdot w + w_q)}$ is the global attention score matrix for the query chunk. The corresponding output of the global attention is computed as:

$$F_{\mathfrak{g}} = A_{\mathfrak{g}} \cdot V, \tag{11}$$

where $F_{\mathfrak{g}} \in \mathbb{R}^{w_q \times d}$ represents the globally updated features for the query chunk. Finally, the updated global representation is passed through the decoding stages, enabling the

model to generate outputs while effectively leveraging information from all chunks.

## 3.2. Parallel Attention Bias

**Theoretical insights into parallel attention bias.** In this section, we develop a theoretical framework to understand *Parallel Attention Bias*, extending the concept of attention collapse (Dong et al., 2021) to parallel attention as described in Section 3.1. We focus on the sparsity behavior of the local attention matrices computed over parallel chunks and examine its impact on both efficiency and accuracy.

**Theorem 3.1.** *Consider the following setup:*

- **Part 1:** *For any $\epsilon > 0$, the sparsity threshold of effective entries in $A_{\mathfrak{l}}^c$ decreases as $w$ increases. $\epsilon$ represents a user-defined threshold controlling sparsity in the attention matrix. As the number of chunks ($C$) increases, $\epsilon$ governs the trade-off between preserving information within each chunk and computational efficiency.*

- **Part 2:** *The number of effective entries $k$ in each row of $A_{\mathfrak{l}}^c$ is upper-bounded by:*

$$k \leq w - \exp\left(O\left(\frac{\log^2(\epsilon \cdot w)}{R^2}\right)\right) \cdot \frac{\delta}{wd},$$

*where $R$ is the rank of the sparse attention matrix, influencing the effective dimensionality of retained attention entries, and $\delta$ is a probability bound controlling the confidence level of the sparsity constraint.*

- **Part 3:** *With high probability $(1 - \delta)$, the number of ineffective entries in each row satisfies:*

$$\lim_{w \to \infty} |\mathcal{S}_\epsilon^{(c)}(A_{\mathfrak{l}}^c[i,:])| = w - k.$$

*Proof Sketch of Theorem 3.1.* **Proof sketch of Part 1:** By utilizing the exponential decay property of local attention weights (as derived in Theorem A.1), the sparsity threshold for effective entries in $A_{\mathfrak{l}}^c$ can be bounded by:

$$\epsilon \geq \exp\left(O(R) \cdot \sqrt{\log(w \cdot (w - k)/\delta)}\right).$$

This inequality indicates that as $w$ increases, the threshold for retaining effective entries becomes stricter, thus limiting the number of such entries.

**Proof sketch of Part 2:** Rearranging the above inequality, we derive an upper bound on $k$, the number of effective entries:

$$k \leq w - \exp\left(O\left(\frac{\log^2(\epsilon \cdot w)}{R^2}\right)\right) \cdot \frac{\delta}{wd}.$$

Thus, the number of effective entries in each row of the attention matrix is $w - k$.

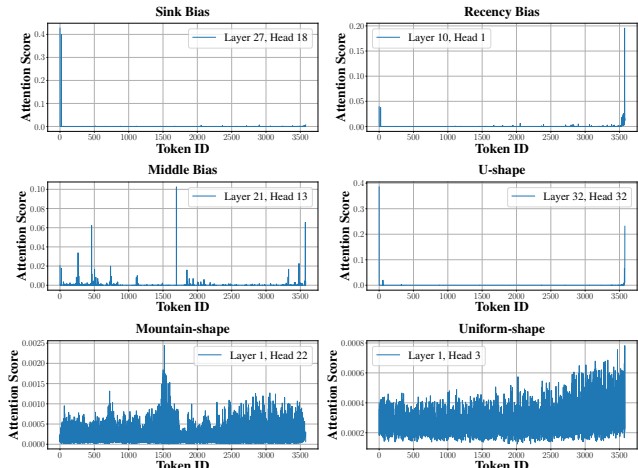

Figure 3: Several types of attention distribution. The Token ID represents the token position in the input text.

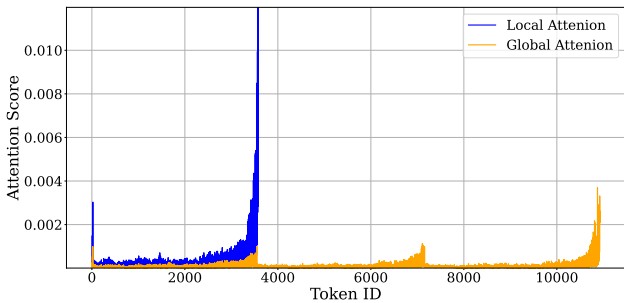

Figure 4: Comparison of local and parallel attention patterns. The blue lines show the local attention distribution within a chunk, while the yellow lines represent the parallel attention patterns in global attention.

**Proof sketch of Part 3:** Substituting the bound on $k$ into the definition of $|\mathcal{S}_\epsilon^{(c)}|$, the number of ineffective entries, we obtain:

$$\lim_{w \to \infty} |\mathcal{S}_\epsilon^c(A_{\mathfrak{l}}^c[i,:])| \geq w - k.$$

Finally, observing that $R = O(\sqrt{\log(w)})$ ensures that the sparsity growth is bounded as $w \to \infty$. A more detailed proof is available in Appendix A. □

**Discussion.** Theorem 3.1 emphasizes the inevitability of attention collapse in parallel attention. If we fix the sparsity threshold $\epsilon$ and keep the number of chunks $C$ constant, as the input sequence length increases, the effective number of attention entries within each chunk decreases as the chunk size $w$ increases, despite partitioning the input sequence into $C$ chunks. The key insights include: i) Each local attention matrix $A_{\mathfrak{l}}^c$ exhibits sparsity behavior akin to the global attention matrix, with most entries becoming negligible for

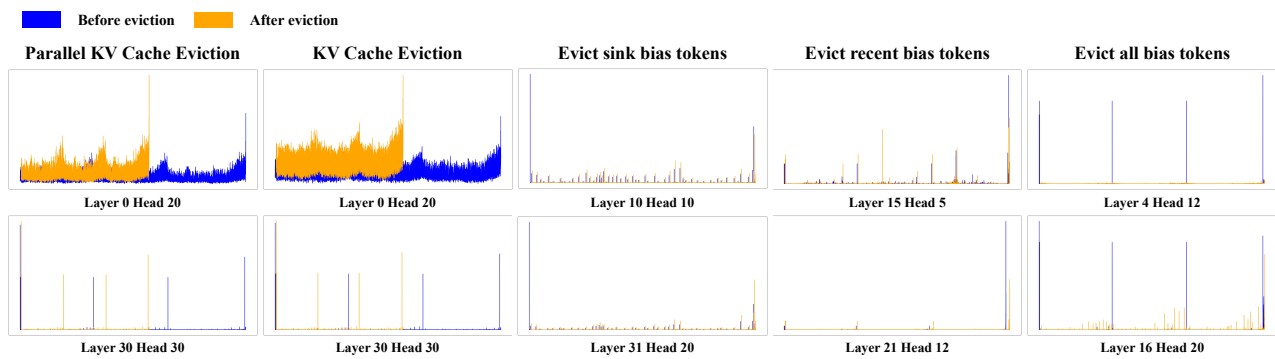

Figure 5: Several types of attention bias and patterns. In the figure, **Parallel KV Cache Eviction** performs independent KV cache eviction within each chunk, while **KV Cache Eviction** unifies this process during global attention. **Parallel KV Cache Eviction** significantly reduces the computational load of global attention.

large $w$. *ii)* When a long sequence is processed in parallel, *attention bias* becomes unavoidable, with the attention mechanism consistently focusing on a small subset of tokens due to its inherent limitations, even when more information is available. Choosing an appropriate sparsity parameter $\epsilon$ can mitigate this issue. *iii)* Dividing the input into chunks reduces computational overhead while preserving sparsity within each chunk, leading to an efficient approximation of global attention.

## 4. Empirical Study of Parallel Attention Bias

In this section, we investigate the attention sink phenomenon in parallel attention and compare its similarities and differences with the regular attention sink phenomenon. Specifically, we explore the following question:

**Q1: What types of attention patterns can be summarized?** In summary, three main types of attention patterns emerge, as illustrated in Figure 3: U-shape, Mountain-shape, and Uniform-shape.

**Observations.** These attention distributions give rise to three corresponding biases: *i)* Attention sink, where focus is concentrated on the initial few tokens. *ii)* Recency bias, where attention is more strongly concentrated at the tail. *iii)* Middle bias, where attention is disproportionately focused on a few tokens in the middle of a sequence. *iv)* These biases manifest in a wavelike pattern, with $R_H$ containing three token types $(R_s, R_m, R_r)$ corresponding to these biases.

**Q2: Is there any difference between the attention bias in parallel attention and the attention bias in classical attention?** In this part, we provide a detailed analysis of bias in parallel attention. We observe in Figure 4 that there are relatively more peaks within the contexts compared to the classic attention mechanism.

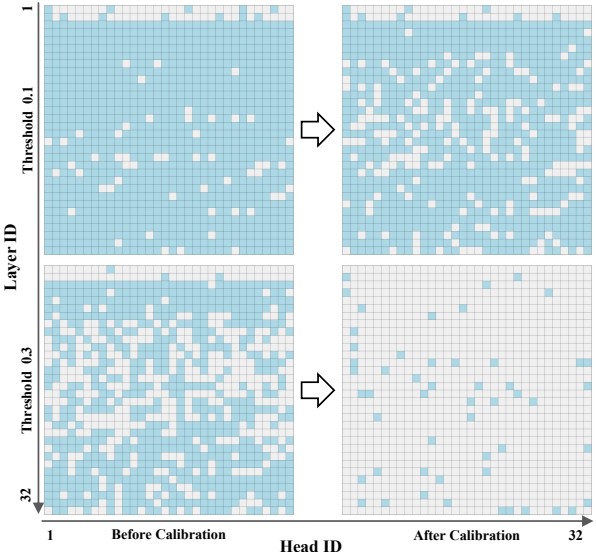

Figure 6: The distribution of tokens with abnormally high attention scores. Blue represents outliers.

**Observations.** *i)* Similar to the blue local attention, the yellow curve shows the U-shaped attention sink repeatedly appearing in global attention. *ii)* Parallel attention and local attention both exhibit severe recency bias, but the bias is significantly mitigated in parallel attention compared to local attention. *iii)* When computing global attention $A_{\mathfrak{g}}$, the model suffers from a more severe recency bias compared to the attention sink, though it is still less pronounced than within $A_{\mathfrak{l}}^c$ (blue line). *iv)* Compared to the classical attention distribution, i.e., the local attention, the peaks of $A_{\mathfrak{g}}$ within the chunk are significantly weakened, indicating that global attention can significantly mitigate recency bias. *In other words, the parallel attention itself can mitigate attention bias.*

| Methods | Single-Document QA | | | Multi-Document QA | | | Summarization | | | Few-shot Learning | | | Synthetic | | Code | | AVG |
|---|---|---|---|---|---|---|---|---|---|---|---|---|---|---|---|---|---|
| | NtrvQA | Qasper | MF-en | HotpotQA | 2WikiMQA | Musique | GovReport | QMSum | MultiNews | TREC | TriviaQA | SAMSum | PCount | PRe | Lcc | RB-P | |
| Max Length | 84123 | 24204 | 17727 | 20325 | 19001 | 20520 | 60515 | 34477 | 16271 | 13049 | 26756 | 21884 | 32699 | 17158 | 37628 | 58822 | 30657 |
| Llama2-7B-chat-hf(4k) | | | | | | | | | | | | | | | | | |
| FullKV(4k) | 18.62 | 19.53 | 35.49 | 31.07 | 26.15 | 9.91 | 25.52 | 20.87 | 26.28 | 62.00 | 82.68 | **40.86** | **5.50** | 10.50 | **61.04** | 55.30 | 33.21 |
| Dynamic-PI | 9.69 | 20.05 | 33.10 | 16.40 | 23.83 | 3.62 | 27.83 | 18.75 | 16.53 | 62.00 | 67.00 | 40.37 | 1.58 | 5.14 | 55.30 | 55.49 | 28.54 |
| NTK-Aware | 13.02 | 14.25 | 31.51 | 29.55 | 30.64 | 11.83 | 28.78 | 16.96 | **26.30** | 62.50 | 74.88 | 39.35 | 4.08 | 4.50 | 49.74 | 49.39 | 30.46 |
| ChunkLlama | 22.97 | 20.52 | 33.71 | 28.91 | 26.14 | 13.84 | 14.84 | 21.62 | 18.13 | 62.50 | 77.15 | 40.83 | 2.03 | 4.00 | 59.81 | 54.33 | 31.33 |
| InfLLM | 18.14 | **22.11** | 29.86 | 30.99 | 30.74 | 9.41 | 26.33 | 20.63 | 26.18 | 62.50 | 84.24 | 39.92 | 3.36 | 6.00 | 60.15 | 55.99 | 32.91 |
| AttenCalibration-NTK | 14.05 | 12.49 | 32.52 | 30.61 | 31.22 | 12.84 | **29.72** | 18.24 | 24.40 | 61.50 | 72.88 | 39.54 | 2.33 | 3.00 | 48.86 | 50.36 | 30.29 |
| Ours | 23.20 | 17.50 | 37.07 | 38.67 | **32.68** | 20.22 | 25.00 | 22.79 | 25.84 | **64.00** | 84.63 | 40.67 | 4.00 | 31.50 | 59.37 | 58.53 | 36.60 |
| Ours-calibration | **24.95** | 19.07 | **38.16** | 39.53 | 32.62 | **22.64** | 25.42 | **22.82** | 26.01 | 63.00 | **85.41** | 40.36 | 5.00 | **32.50** | 59.04 | **58.84** | **37.21** |
| Ours-compression | 23.32 | 16.97 | 35.25 | 39.49 | 32.47 | 20.17 | 24.33 | 21.97 | 25.68 | 63.50 | 84.46 | 40.81 | 4.00 | 31.50 | 59.43 | 58.54 | 36.37 |
| Ours-calibration-compression | 24.04 | 18.39 | 38.03 | **39.89** | 35.38 | 22.15 | 24.26 | 22.46 | 24.51 | 63.50 | 84.83 | 40.73 | 4.00 | 31.50 | 57.67 | 58.48 | 36.86 |
| Llama3-8B-instruct(8k) | | | | | | | | | | | | | | | | | |
| FullKV(8k) | 24.31 | 38.13 | 39.69 | 44.16 | 35.66 | 21.00 | 28.35 | 23.06 | **26.96** | 73.00 | 90.13 | 42.46 | 4.61 | 68.50 | 60.46 | **56.11** | 42.29 |
| Dynamic-PI | 21.71 | 36.66 | 38.24 | 33.70 | 35.48 | 14.28 | 29.41 | 22.04 | 25.55 | 74.50 | 82.61 | **42.62** | 2.33 | 85.59 | 58.22 | 47.16 | 40.63 |
| NTK-Aware | 25.92 | 37.54 | 42.23 | 48.32 | 36.96 | 27.51 | 33.74 | 24.13 | 26.35 | 50.50 | 88.84 | 42.53 | 7.24 | **95.61** | 34.84 | 39.04 | 41.33 |
| ChunkLlama | 25.01 | 37.39 | **43.52** | 49.37 | 37.56 | **30.95** | 17.57 | 23.51 | 19.72 | **76.00** | 90.38 | 42.44 | 4.71 | 67.95 | **61.10** | 52.57 | 42.47 |
| InfLLM | 19.93 | **43.52** | 40.58 | 48.31 | 35.99 | 23.25 | 30.49 | 21.60 | 26.53 | 74.00 | 90.93 | 42.30 | 8.00 | 74.00 | 58.98 | 52.46 | 43.18 |
| AttenCalibration-NTK | 26.54 | 37.52 | 41.13 | 47.56 | 38.98 | 26.51 | **34.21** | 23.35 | 25.64 | 45.50 | 89.23 | 42.21 | 4.81 | 93.51 | 36.86 | 42.82 | 41.02 |
| Ours | 26.67 | 39.05 | 42.66 | 49.58 | 40.02 | 26.23 | 29.10 | 24.18 | 26.74 | 69.00 | **91.03** | 42.07 | 7.81 | 92.38 | 58.06 | 53.54 | 44.93 |
| Ours-calibration | **26.89** | 39.46 | 42.01 | **49.88** | **41.41** | 26.68 | 29.17 | 24.55 | 26.77 | 72.50 | 90.53 | 42.13 | **8.02** | 92.75 | 58.06 | 53.97 | **45.21** |
| Ours-compression | 26.18 | 36.56 | 39.72 | 47.10 | 34.89 | 24.96 | 27.03 | 23.86 | 24.52 | 67.00 | 89.55 | 41.20 | 7.37 | 92.29 | 58.51 | 52.15 | 43.31 |
| Ours-calibration-compression | 26.46 | 37.49 | 41.28 | 48.28 | 36.29 | 26.68 | 26.79 | **24.98** | 25.18 | 69.00 | 90.37 | 40.72 | 7.34 | 91.29 | 57.30 | 53.97 | 44.31 |

Table 1: Length Extrapolation Performance Comparison across Different Tasks. **Ours-calibration** and **Ours-compression** both represent parallel KV Cache Eviction, where the former evicts tokens of $R_h$, and the latter evicts tokens of $R_l$. **Ours-calibration-compression** represents the simultaneous adoption of both eviction strategies. **FullKV** refers to truncating the context to 4k or 8k lengths (without extrapolation) for generation.

**Q3: Can the eviction strategy alleviate attention bias?** By evicting different types of $R_H$ at different layers, we have the following observations:

**Observations.** *i)*: From Figure 5, we can find that KV cache eviction exacerbates the bias. However, *parallel KV cache eviction* can achieve a more stable distribution. *ii)*: Evicting sink bias tokens in the early layers may exacerbate attention bias, but evicting them in the deeper layers can mitigate this attention bias. *iii)*: Evicting recency bias tokens in the intermediate layers can mitigate attention bias, while evicting recency bias tokens in the deeper layers redistributes the attention scores obtained by the recency bias tokens to the intermediate tokens. *iv)*: Simultaneously evicting sink bias and recency bias tokens can alleviate attention bias in the intermediate layers (Layer 16). *v)*: As shown in Figure 6, evicting tokens with abnormally high attention scores appears to effectively mitigate attention bias within the model. However, the impact of this strategy on task-specific performance remains uncertain. We will investigate this further in our experiments.

## 5. Experiment

### 5.1. Experimental Settings

**Models, Baselines, and Tasks.** We compare our method with existing length extrapolation approaches, including Position Interpolation (PI) (Chen et al., 2023b), NTK-Aware (LocalLLaMA, 2023b), ChunkLlama (An et al., 2024), AttenCalibration (Yu et al., 2024), APE (Yang et al., 2025b), StarAttention (Acharya et al., 2024), and

InfLLM (Xiao et al., 2024), on LongBench (Bai et al., 2023) and InfiniteBench (Zhang et al., 2024), evaluating them on Llama2-7B-chat-hf (Touvron et al., 2023), LLaMA3.1 (Grattafiori et al., 2024), Qwen2.5 (Yang et al., 2025a) and Llama-3-8B-Instruction (AI, 2024). We also compare our method with the following open-source and closed-source models trained on long-context data: ChatGLM-3-6B-128K (GLM et al., 2024), Kimi-Chat (AI, 2023), Yi-6B-200K (01.AI, 2023a), Yi-34B-200K (01.AI, 2023b), Claude-2 (Anthropic, 2023), Yarn-Mistral-7b-128k (Peng et al., 2023), and GPT-4 (Achiam et al., 2023). Since AttenCalibration only calibrates the attention distribution and lacks the capability for length extrapolation, we incorporate NTK-aware techniques to enable this functionality, resulting in AttenCalibration-NTK. Details of our hyperparameters are provided in Appendix C.

### 5.2. Length Extrapolation Settings

**Main results.** We present our method in Table 1, showing the performance of several strong baselines on LongBench. We have the following main findings: *i)*: Our method is the *only one* that surpasses FullKV (i.e., the baseline without any length extrapolation) across different backbones. *ii)*: Section 4 reveals that parallel KV cache compression exacerbates attention bias. However, combining it with the eviction $R_H$ method to calibrate the attention distribution, i.e., Ours-calibration-compression, can restore the performance to that of the original KV cache size. *iii)*: Chunk-based length extrapolation methods, such as InfLLM and ChunkL-

| Llama2-7B-chat-hf(4k) | | | | | | | |
| --- | --- | --- | --- | --- | --- | --- | --- |
| **Methods** | **R.PK** | **R.Num** | **R.KV** | **En.MC** | **Math.F** | **Code.Debug** | **Average** |
| Max Length | 125k | 125k | 175k | 834k | 120k | 258k | 273k |
| FullKV | 1.36 | 1.86 | 0.4 | 0.44 | 17.43 | 21.57 | 7.18 |
| Dynamic-PI | 0.17 | 0.00 | 0.00 | 7.42 | 2.00 | 21.32 | 5.15 |
| NTK-Aware | 2.54 | 0.00 | 0.00 | 3.06 | 7.71 | 18.78 | 5.35 |
| ChunkLlama | 12.88 | 13.22 | 0.20 | 0.87 | 17.14 | 22.08 | 11.07 |
| InfLLM | 100.00 | 96.61 | 2.40 | 29.80 | 16.86 | 22.34 | 44.67 |
| AttenCalibration-NTK | 0.00 | 0.00 | 0.00 | 1.06 | 5.71 | 19.24 | 4.34 |
| Ours | 100.00 | 97.63 | 20.60 | 33.62 | 19.71 | 25.13 | 49.45 |
| Ours-calibration | 100.00 | 98.64 | 22.80 | 36.24 | 19.71 | 30.20 | **51.27** |
| Ours-compression | 97.80 | 87.96 | 5.00 | 35.81 | 15.86 | 27.41 | 44.97 |
| Ours-calibration-compression | 97.97 | 90.14 | 10.80 | 35.46 | 15.86 | 28.21 | 46.41 |
| Llama3-8B-instruct(8k) | | | | | | | |
| **Methods** | **R.PK** | **R.Num** | **R.KV** | **En.MC** | **Math.F** | **Code.Debug** | **Average** |
| FullKV | 6.10 | 6.27 | 4.80 | 42.79 | 38.57 | 22.34 | 20.15 |
| Dynamic-PI | 0.00 | 0.00 | 0.00 | 28.82 | 29.71 | **24.62** | 13.86 |
| NTK-Aware | 3.39 | 8.47 | 9.40 | 35.37 | 39.43 | 17.77 | 18.97 |
| ChunkLlama | 3.05 | 9.15 | 3.60 | 13.54 | 34.29 | 11.42 | 12.51 |
| AttenCalibration-NTK | 4.58 | 8.47 | 12.40 | 34.28 | 36.57 | 22.68 | 19.83 |
| InfLLM | 100.00 | 99.00 | 5.00 | 43.70 | 23.70 | 22.08 | 48.91 |
| Ours | 100.00 | 99.83 | 92.80 | 54.59 | 40.00 | 22.84 | 68.34 |
| Ours-calibration | 100.00 | 99.49 | **93.80** | 56.77 | 40.00 | 23.24 | **68.88** |
| Ours-compression | 100.00 | 99.83 | 89.20 | 55.48 | 40.00 | 21.32 | 67.64 |
| Ours-calibration-compression | 100.00 | 99.83 | 91.00 | 56.77 | 40.00 | 22.20 | 68.30 |
| Other proprietary models | | | | | | | |
| **Models** | **R.PK** | **R.Num** | **R.KV** | **En.MC** | **Math.F** | **Code.Debug** | **Average** |
| GPT-4 | 100.00 | 100.00 | 89.00 | 67.25 | 60.00 | **37.06** | **75.55** |
| Kimi-Chat | 98.14 | 95.42 | 53.60 | **72.49** | 12.57 | 17.14 | 58.23 |
| Claude-2 | 97.8 | 98.14 | 65.40 | 62.88 | 32.29 | 17.77 | 62.38 |
| Other open-source models | | | | | | | |
| **Models** | **R.PK** | **R.Num** | **R.KV** | **En.MC** | **Math.F** | **Code.Debug** | **Average** |
| YaRN-Mistral-7B-128k | 92.71 | 56.61 | <5 | 27.95 | **17.14** | **60.00** | 42.82 |
| Yi-6B-200K | 100 | 94.92 | <5 | 36.68 | <5 | <5 | 39.85 |
| Yi-34B-200K | 100 | 100 | <5 | **38.43** | <5 | 25.71 | **44.86** |
| ChatGLM-3-6B-128K | 92.2 | 80.68 | <5 | 10.48 | <5 | 7.71 | 32.68 |

Table 2: The model's performance on the InfiniteBench dataset across different datasets.

| Llama2-7B-chat-hf(4k) | | | | | | | |
| --- | --- | --- | --- | --- | --- | --- | --- |
| **Methods** | **2k** | **4k** | **8k** | **16k** | **32k** | **64k** | **128k** |
| Llama2-7b | 7.03 | 6.71 | $> 10^2$ | $> 10^2$ | $> 10^2$ | $> 10^2$ | $> 10^2$ |
| Dynamic-PI | 7.03 | 6.71 | 7.02 | 11.62 | 59.31 | $> 10^2$ | $> 10^2$ |
| NTK-Aware | 8.61 | 8.41 | 8.29 | 7.19 | 40.71 | $> 10^2$ | $> 10^2$ |
| ChunkLlama | **7.03** | **6.71** | **6.42** | **5.01** | **4.82** | 12.36 | 43.57 |
| InfLLM | 23.24 | 23.46 | 21.86 | 20.40 | 19.84 | 18.26 | 18.97 |
| Ours | 8.01 | 9.71 | 11.97 | 10.46 | 11.34 | **11.58** | **12.56** |
| Llama3-8B-instruct(8k) | | | | | | | |
| **Methods** | **2k** | **4k** | **8k** | **16k** | **32k** | **64k** | **128k** |
| Llama3-8b | 9.90 | 9.15 | 7.94 | 63.13 | $> 10^2$ | $> 10^2$ | $> 10^2$ |
| Dynamic-PI | 9.90 | 9.15 | 17.25 | 69.96 | $> 10^2$ | $> 10^2$ | $> 10^2$ |
| NTK-Aware | 10.71 | 9.66 | 8.16 | 6.74 | 8.06 | 77.63 | $> 10^2$ |
| ChunkLlama | 9.88 | 9.14 | 7.92 | 6.57 | 6.13 | 5.33 | **5.40** |
| InfLLM | 8.50 | 9.30 | 8.72 | 9.47 | 8.98 | 9.66 | 9.10 |
| Ours | **5.85** | **6.75** | **6.65** | **6.30** | **5.61** | **5.13** | 5.72 |

Table 3: We test the perplexity on the NarrativeQA (Kočiský et al., 2018) test set.

lama, generally perform better than position encoding-based methods such as Dynamic-PI and NTK-Aware. *iv)*: Directly calibrating the attention distribution in NTK-aware length extrapolation methods, such as *AttenCalibration-NTK*, leads to strong performance primarily on the longest datasets, including NtrvQA, GovReport, and RB-P. This suggests that the effect of attention distribution calibration becomes increasingly significant as input length grows.

**Extrapolating beyond 128K context lengths.** We evaluate the performance under extremely long contexts in Table 2, comparing it with several powerful open-source and closed-source models. These models are trained on context lengths exceeding 128K, and thus do not require additional extrapolation capabilities to handle ultra-long contexts. We have the following findings: *i)*: Our method performs excep-

tionally well on needle-in-a-haystack retrieval tasks (R.PK, R.Num, R.KV), being the *only model* capable of achieving over 90% accuracy across all tasks, surpassing even the strongest closed-source model, GPT-4. *ii)*: Position encoding-based length extrapolation methods, such as NTK-Aware, Dynamic-PI, generally struggle to achieve good performance on tasks with ultra-long contexts compared to chunk-based extrapolation approaches. *iii)*: Our training-free extrapolation method, using an 8K window, is the *only approach* that surpasses the powerful closed-source models Kimi-Chat and Claude-2, achieving 91.17% of GPT-4's performance on ultra-long contexts with an 8B model.

**Language modeling.** To further compare the performance of our method in language modeling, we present the results of perplexity (PPL) calculations on the NarrativeQA test set in Table 3, which reflect the model's performance in long-context language modeling. For fair comparison, we typically calculate the PPL for the query chunk, as it corresponds to the model's decoding phase. *i)*: Chunk-based position extrapolation methods (ChunkLlama, InfLLM, and Ours) achieve significantly lower PPL compared to position encoding-based methods (Dynamic-PI and NTK-Aware). *ii)*: Position encoding-based methods start to collapse in performance for language modeling when the length exceeds 32k. *iii)*: As the number of chunks increases (from 2K to 128K), our method still demonstrates consistent perplexity stability across different lengths. Surprisingly, ChunkLlama maintains high performance on Llama3-8B-instruct, outperforming other methods.

### 5.3. Evaluation in Long-context Models

To demonstrate the effectiveness of our method for long-context models, we evaluate it on two models trained with contexts up to 128K, which therefore do not require any extrapolation capabilities: LLaMA3.1 (Grattafiori et al., 2024) and Qwen2.5 (Yang et al., 2025a), used as base models. All evaluations are conducted under a consistent 24K KV cache size. While standard baselines directly utilize the full 24K position encodings, our method applies extrapolation techniques to reuse a 6K position encoding, thereby supporting longer input lengths without modifying the model architecture. To ensure fairness, we standardize the use of special prompt tokens across models (e.g., `<|begin_of_text|>` for LLaMA3.1), which we observe to have a significant impact on performance. All reported results correspond to configurations that include these prompt tokens.

As shown in Table 4 and Table 5, our PARALLELCOMP method consistently improves performance across a broad range of tasks. On LongBench, we observe modest but consistent gains compared to baselines. On InfiniteBench, which emphasizes ultral-long context understanding, our method demonstrates significant improvements, particularly

| Method | NARR | QAS | MULT | HOPT | 2WKI | MUS | GOV | QMS | NEWS | TREC | TRIV | SSM | PCNNT | PREN | LCC | REP | AVG |
|---|---|---|---|---|---|---|---|---|---|---|---|---|---|---|---|---|---|
| Llama3.1 | 27.80 | 44.25 | 49.46 | 47.86 | 40.54 | 23.64 | 32.64 | 22.90 | 26.90 | 38.00 | 88.44 | 25.64 | 2.02 | 92.00 | 10.35 | 18.64 | 36.94 |
| ParallelComp-Llama3.1 | **29.45** | **45.98** | **50.67** | 48.36 | **46.56** | 23.32 | 32.60 | **24.29** | **27.34** | 38.50 | 86.72 | 25.93 | 0.05 | 95.00 | 14.15 | **21.42** | 38.15 |
| Qwen2.5 | 27.83 | 41.31 | 50.41 | 53.52 | 44.68 | **30.00** | 33.38 | 24.01 | 25.40 | 71.00 | 86.10 | 39.91 | **7.25** | **100.00** | 6.86 | 7.88 | 40.60 |
| ParallelComp-Qwen2.5 | 28.42 | 42.24 | 50.54 | **56.26** | 42.02 | 28.25 | **33.43** | 23.20 | 25.20 | **71.50** | **89.21** | **41.84** | 5.00 | 93.50 | **20.73** | 13.34 | **41.54** |

Table 4: Performance on LongBench benchmark. Models are evaluated under a 24K KV cache budget.

| Method | PS | NUM | KV | EN.MC | MATH | CODE | AVG |
|---|---|---|---|---|---|---|---|
| Llama3.1 | 5.59 | 26.25 | 18.60 | 32.86 | 31.52 | 22.56 | 26.36 |
| ParallelComp-Llama3.1 | **100.00** | **83.56** | **88.60** | 66.38 | 37.14 | 22.08 | 59.55 |
| Qwen2.5 | 59.32 | 58.31 | 33.80 | 61.39 | 85.71 | 23.76 | 53.72 |
| ParallelComp-Qwen2.5 | **100.00** | 76.27 | 63.40 | **66.86** | **92.57** | **24.75** | **70.64** |

Table 5: Performance on InfiniteBench with different models. Models are evaluated under a 24K KV cache budget.

for tasks such as PS, NUM, and KV. *These results indicate that extrapolated position encodings can be effectively reused in parts of the model's position encodings to extend context length, achieving comparable or even improved performance relative to using full position encodings in long-context models.*

| | Llama2-7B-chat-hf(4k) | | | | | | |
|---|---|---|---|---|---|---|---|
| **Methods** | **R.PK** | **R.Num** | **R.KV** | **En.MC** | **Math.F** | **Code.Debug** | **Average** |
| Max Length | 125k | 125k | 175k | 834k | 120k | 258k | 273k |
| Ours | **100.00** | 97.63 | 20.60 | 33.62 | **19.71** | 25.13 | 49.45 |
| Ours-calibration | **100.00** | **98.64** | **22.80** | 36.24 | **19.71** | 30.20 | **51.27** |
| Sink-eviction-layer-1-8 | 99.32 | 42.71 | 2.20 | 37.12 | 17.71 | 22.84 | 36.98 |
| Sink-eviction-layer-9-16 | 100.00 | 91.19 | 11.00 | 37.12 | 14.86 | 24.37 | 46.42 |
| Sink-eviction-layer-17-23 | 100.00 | 97.80 | 20.80 | 33.19 | 19.14 | 30.96 | 50.32 |
| Sink-eviction-layer-24-31 | 100.00 | 97.63 | 20.20 | 31.88 | 18.00 | 29.19 | 49.48 |
| Recency-eviction-layer-1-8 | 100.00 | 96.44 | 2.60 | 33.19 | 16.00 | 19.54 | 44.63 |
| Recency-eviction-layer-9-16 | 100.00 | 97.80 | 15.80 | **37.99** | 10.86 | 23.10 | 47.59 |
| Recency-eviction-layer-17-23 | 100.00 | 97.97 | 20.40 | 23.58 | 16.00 | **32.74** | 48.45 |
| Recency-eviction-layer-24-31 | 100.00 | 97.63 | 20.60 | 35.81 | 18.57 | 25.89 | 49.75 |
| Middle-eviction-layer-1-8 | 100.00 | 97.29 | 20.60 | 34.93 | 18.29 | 22.84 | 48.99 |
| Middle-eviction-layer-9-16 | 100.00 | 97.63 | 20.60 | 33.62 | 16.00 | 26.40 | 49.04 |
| Middle-eviction-layer-17-23 | 98.81 | 97.46 | 20.00 | 34.06 | 19.14 | 30.20 | 49.95 |
| Middle-eviction-layer-24-31 | 100.00 | 97.46 | 19.80 | 30.13 | 19.43 | 28.93 | 49.29 |

Table 6: Ablation of Llama2-7B-chat-hf on InfiniteBench. Ours-calibration refers to the approach where layers 9-16 adopt the recency bias token eviction method, while layers 25-32 evict sink bias tokens, and layers 1-8 evict middle bias tokens. Other methods follow the naming format [Evicted Token Type]-eviction-layer-[Evicted Layer Range].

### 5.4. Ablation of Attention Bias

We present in Table 6 the impact of evicting different bias tokens at various layers on different tasks. We have the following observations: *i):* The $R_s$ in the shallow layers (1-8) is crucial for retrieval tasks. Without these tokens, the model's performance will be significantly impaired. *ii):* The $R_r$ in the deeper layers (layers 9-16) plays a crucial role in the model's reasoning abilities. Evicting these tokens results in a decline in performance on coding and math tasks. *iii):* Shallow $R_m$ (layers 1-8) damages the model's understanding ability, and evicting them can improve the model's performance. Deep $R_m$ (layers 24-31) contributes to the model's ability in reading comprehension tasks (En.MC), and evicting them harms the model's performance. *iv):* $R_r$ in the early layers (layers 1-8) is important for the model's in-context learning ability. For a detailed analysis of this

phenomenon, please refer to Appendix B.

| Chunk Number | Ours | | Ours-Compression | |
|---|---|---|---|---|
| | Prefill (s) / Generation (ms/token) | Max Memory Used (MB) | Prefill (s) / Generation (ms/token) | Max Memory Used (MB) |
| 1 | 1317.72 / 24.30 | 19394 | 1317.72 / 22.16 | 16994 |
| 4 | 321.40 / 43.69 | 35518 | 321.40 / 23.28 | 24734 |
| 8 | 160.70 / 72.53 | 47758 | 160.70 / 31.21 | 36396 |
| 12 | 111.67 / 102.94 | 65980 | 111.67 / 39.61 | 48458 |
| 16 | N/A | Out-of-Memory | 82.36 / 49.25 | 59140 |
| 20 | N/A | Out-of-Memory | 65.23 / 56.25 | 71302 |
| 23 | N/A | Out-of-Memory | 56.07 / 57.19 | 79742 |
| 24 | N/A | Out-of-Memory | N/A | Out-of-Memory |

Table 7: Throughput analysis. We evaluate on Llama2-7B-chat-hf and compare the improvement in chunk throughput with the use of parallel KV cache compression. Time tests were performed on the NarrativeQA dataset. Experiments are conducted on an AMD Instinct MI210 64GB GPU.

### 5.5. Throughput Analysis

We mainly focus on the throughput of chunks during context parallelism. Therefore, we compare the maximum number of parallel chunks and the memory usage before and after parallel KV cache compression. Table 7 presents the memory usage of the model using the parallel KV cache eviction strategy. On a single GPU, by compressing the KV cache size of each chunk to half of its original size, we achieve a 1.76x improvement in chunk throughput, thereby achieving a 23.50x acceleration in the prefill stage with negligible performance loss.

## 6. Conclusion

In this paper, we propose PARALLELCOMP, a training-free and parallel long-context compression framework that significantly enhances the extrapolation capability of large language models (LLMs) for ultra-long contexts. PARALLELCOMP overcomes the critical memory bottlenecks in length extrapolation and systematically analyzes the attention bias that arises in such settings. Specifically, our method allows 8B LLMs to extend inference length from 4K to 128K tokens on a single A100 80GB GPU without retraining or significant degradation in performance. By leveraging chunk-based parallel attention, dynamic KV cache eviction, and an attention calibration strategy, our approach alleviates both excessive memory usage and the attention sink phenomenon. Extensive theoretical and empirical results demonstrate that PARALLELCOMP effectively mitigates attention bias and enables robust, end-to-end inference. Notably, our method achieves $91.17\%$ of GPT-4's performance on ultra-long context tasks using an 8B model, outperforming various state-of-the-art closed-source models. These findings pave the way for more scalable and efficient long-context inference.

## Impact Statement

This paper presents work whose goal is to advance the field of Machine Learning. There are many potential societal consequences of our work, none which we feel must be specifically highlighted here.

## Acknowledgements

This work is supported in part by the Theme-based Research Scheme (TRS) project T45-701/22-R of the Research Grants Council (RGC), Hong Kong SAR, and in part by the AVNET-HKU Emerging Microelectronics & Ubiquitous Systems (EMUS) Lab.

## Future Work

While PARALLELCOMP enables efficient length extrapolation up to 128K tokens, future work can explore its application to long-chain reasoning and ultra-long in-context learning (Xiong et al., 2023). This includes designing methods for reasoning over 100K+ tokens, developing test-time scaling. Further exploration into in-context learning will also be critical for reliable deployment in real-world long-context reasoning.

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

# A. Proofs for Parallel Attention Bias

## A.1. Exponential Decay of Local Attention

We formalize the exponential decay of attention values as a function of relative distance when using relative position encodings (Su et al., 2024). This behavior is modulated by a sparsity control parameter $R$, which regulates the rate at which attention mass diminishes over increasing distance.

**Theorem A.1** (Exponential Decay of Attention). *Let $A_{\mathfrak{l}}^c[i, j]$ denote the local attention weight from token $i$ to token $j$ within chunk $c$, computed using scaled dot-product attention with relative position encoding. Suppose the dot product between query $q_i$ and key $k_j$ depends linearly on their relative distance, i.e.,*

$$q_i \cdot k_j \approx -\alpha \cdot d(i, j),$$

*where $d(i, j)$ is the positional distance and $\alpha > 0$ is a constant. Then the attention value exhibits exponential decay:*

$$A_{\mathfrak{l}}^c[i, j] \sim \exp\left(-O(R) \cdot d(i, j)\right),$$

*where $R$ is a sparsity parameter controlling the effective attention range.*

*Proof.* The local attention matrix is given by

$$A_{\mathfrak{l}}^c[i, j] = \frac{\exp\left(\frac{q_i \cdot k_j}{\sqrt{d}}\right)}{\sum_{j'=1}^{w} \exp\left(\frac{q_i \cdot k_{j'}}{\sqrt{d}}\right)}.$$

Assuming $q_i \cdot k_j \approx -\alpha \cdot \mathrm{d}(i, j)$, we obtain:

$$\exp\left(\frac{q_i \cdot k_j}{\sqrt{d}}\right) \approx \exp\left(-\frac{\alpha}{\sqrt{d}} \cdot \mathrm{d}(i, j)\right).$$

Letting $O(R) := \alpha/\sqrt{d}$ leads to:

$$A_{\mathfrak{l}}^c[i, j] \propto \exp\left(-O(R) \cdot \mathrm{d}(i, j)\right),$$

establishing the exponential decay behavior. $\square$

**Discussion.** This exponential decay arises from the softmax mechanism and relative position encoding: as the relative distance increases, the inner product diminishes, leading to exponentially suppressed attention values. The parameter $R$ determines how rapidly this suppression occurs. A higher $R$ yields slower decay (broader attention), while a lower $R$ enforces sharper locality—crucial for analyzing sparsity and receptive fields in attention models.

**Theorem A.2** (Tail Bound on Effective Attention Mass). *Suppose $A_{\mathfrak{l}}^c[i, j] \sim \exp(-O(R) \cdot d(i, j))$ with $R = O(\sqrt{\log w})$. Let $\epsilon > 0$ be a threshold and $k$ be the number of ineffective entries in row $i$ of the local attention matrix, i.e., those satisfying $A_{\mathfrak{l}}^c[i, j] \leq \epsilon$. Then for any $0 < \delta < 1$,*

$$\epsilon \geq \exp\left(-O(R) \cdot \sqrt{\log\left(\frac{w(w-k)}{\delta}\right)}\right).$$

*Proof.* Let us consider the attention weight between query $q_i$ and key $k_j$ (within chunk $c$) given by

$$A_{\mathfrak{l}}^c[i, j] = \frac{\exp\left(\frac{q_i \cdot k_j}{\sqrt{d}}\right)}{\sum_{j'=1}^{w} \exp\left(\frac{q_i \cdot k_{j'}}{\sqrt{d}}\right)}.$$

Assume the dot product $q_i \cdot k_j$ (after position encoding) satisfies

$$q_i \cdot k_j = -\alpha \cdot d(i, j) + Z_{i,j},$$

where $d(i, j)$ is the relative distance, according to Theorem A.1, $\alpha > 0$ is a scaling constant, and $Z_{i,j}$ is a mean-zero, sub-Gaussian noise term arising from randomness in $q_i$ and $k_j$.

**Sub-Gaussian concentration.** Suppose $Z_{i,j}$ is sub-Gaussian with parameter $\sigma^2$ (variance proxy). Since $q_i, k_j \in \mathbb{R}^d$ are assumed to be independent and isotropic (or sufficiently well-behaved), the inner product $Z_{i,j}$ scales as $\sim \mathcal{N}(0, d)$, so after normalization by $\sqrt{d}$, we have:

$$\frac{Z_{i,j}}{\sqrt{d}} \sim \mathcal{N}(0, 1)$$

and more generally, sub-Gaussian(1) for all entries.

**Tail probability for an entry exceeding $\epsilon$.** We are interested in the probability a particular attention entry $A_{\mathfrak{l}}^c[i, j]$ exceeds a fixed threshold $\epsilon > 0$:

$$\mathbb{P}\left(A_{\mathfrak{l}}^c[i, j] > \epsilon\right).$$

The numerator is $\exp\left(\frac{q_i \cdot k_j}{\sqrt{d}}\right)$, which, via the deterministic and random parts, is $\exp\left(-\frac{\alpha}{\sqrt{d}} \mathrm{d}(i, j) + \frac{Z_{i,j}}{\sqrt{d}}\right)$. The denominator $Z_i$ is the sum over all $w$ such terms.

By the softmax property, after normalization, it holds that for entries with large $d(i, j)$ (i.e., far from the diagonal), the probability concentrates at low values, and for an entry to exceed $\epsilon$ the exponent must be large:

$$\exp\left(-\frac{\alpha}{\sqrt{d}} d(i, j) + \frac{Z_{i,j}}{\sqrt{d}}\right) > \epsilon Z_i$$

or

$$-\frac{\alpha}{\sqrt{d}} d(i, j) + \frac{Z_{i,j}}{\sqrt{d}} > \log(\epsilon Z_i).$$

**Union bound over all entries.** We need to ensure with high probability (at least $1 - \delta$) that at most $k$ entries per row (out of $w$) exceed $\epsilon$. That is, we want (by union bound)

$$w(w - k) \cdot \mathbb{P}\left(A_{\mathfrak{l}}^c[i, j] > \epsilon\right) \le \delta.$$

Then,

$$\mathbb{P}\left(A_{\mathfrak{l}}^c[i, j] > \epsilon\right) \le \frac{\delta}{w(w - k)}$$

which, after inverting the softmax and using sub-Gaussian tail estimates for $Z_{i,j}$, gives

$$\mathbb{P}\left(\frac{Z_{i,j}}{\sqrt{d}} > \log(\epsilon Z_i) + \frac{\alpha}{\sqrt{d}} \mathrm{d}(i, j)\right) \le \frac{\delta}{w(w - k)}.$$

The sub-Gaussian tail bound reads for $t > 0$:

$$\mathbb{P}(X > t) \le \exp(-t^2/2).$$

Setting $t^2/2 = \log\left(\frac{w(w-k)}{\delta}\right)$ leads to

$$t = \sqrt{2 \log\left(\frac{w(w - k)}{\delta}\right)}$$

and relating $t$ back to normalization and decay constants, finally yields

$$\epsilon \gtrsim \exp\left(-O(R) \cdot \sqrt{\log\left(\frac{w(w - k)}{\delta}\right)}\right)$$

where $R = \alpha/\sqrt{d}$ (showing dependence on $d$).

**The role of $d$ in the bound.** The normalization by $\sqrt{d}$ plays a key role: as $d$ increases, the effective noise in the dot product becomes less significant relative to the deterministic decay $-\alpha d(i, j)$, pulling most attention values closer to zero except for a small set (nearby positions). This effect concentrates the attention and tightens the tail, making large attention far from the diagonal even less probable.

Note that, in total union bound, we consider $w^2$ possible entries and sometimes further divide by $d$, reflecting the increased concentration with high dimension or the independence among $d$ channels (as found in certain information-theoretic arguments for high-dimensional softmax). This dependence enters the denominator for the upper bound on $k$:

$$k \leq w - \exp\left( O\left( \frac{\log^2(\epsilon w)}{R^2} \right) \right) \frac{\delta}{wd}.$$

**Summary.** The dependence on $d$ in the denominator comes from sub-Gaussian concentration for scalar products in dimension $d$. As $d$ grows, the likelihood of far-off entries (large distance) carrying significant attention decays rapidly due to stronger concentration of measure, resulting in increased sparsity of the attention matrix for fixed threshold $\epsilon$. Therefore, for sufficiently large $w$ and $d$, with fixed $\epsilon$,

$$k \ll w,$$

implying

$$\lim_{w \to \infty} |\mathcal{S}_\epsilon^{(c)}(A_{\mathfrak{l}}^c[i, :])| = w - k,$$

where $k = o(w)$.

$\square$

**Remark.** The appearance of the term $d$ in the denominator reflects how higher model dimensionality increases the concentration of dot products, making rare large excursions (that could create significant attention values far from the main focus) even less probable. Thus, the sparsity bound becomes stricter as $d$ increases. This probabilistic tail behavior is essential for understanding how the structure and dimension of the attention model affect the sparsity and locality of its learned representations.

### A.2. Sparsity in Local Attention

**Theorem A.3** (Sparsity of Local Attention). *Let $X \in \mathbb{R}^{N \times d}$ be partitioned into $C$ chunks of width $w$. For each chunk $c$, let local attention matrix $A_{\mathfrak{l}}^c \in \mathbb{R}^{w \times w}$ be computed via*

$$A_{\mathfrak{l}}^c = \text{Softmax}\left( \frac{Q^c K^{c\top}}{\sqrt{d}} \right).$$

*Let $\mathcal{S}_\epsilon^{(c)}(A_{\mathfrak{l}}^c[i, :]) = \{j \in [w] \mid A_{\mathfrak{l}}^c[i, j] > \epsilon\}$ for fixed $\epsilon > 0$, and set $R = O(\sqrt{\log w})$.*

*Then for any $0 < \delta < 1$, as $w \to \infty$, with high probability $1 - \delta$,*

$$\left| \mathcal{S}_\epsilon^{(c)}(A_{\mathfrak{l}}^c[i, :]) \right| = w - k,$$

*where $k = o(w)$ depends on $\epsilon$ and $R$.*

*Proof.* We begin with Theorem A.2, which provides a tail bound for the effective attention mass:

$$\epsilon \geq \exp\left( -O(R) \cdot \sqrt{\log\left( \frac{w(w - k)}{\delta} \right)} \right).$$

Taking logarithms on both sides gives

$$\log \epsilon \geq -O(R) \cdot \sqrt{\log\left( \frac{w(w - k)}{\delta} \right)}.$$

Rearranging, we have

$$\sqrt{\log\left(\frac{w(w-k)}{\delta}\right)} \leq \frac{|\log\epsilon|}{O(R)},$$

so

$$\log\left(\frac{w(w-k)}{\delta}\right) \leq \frac{\log^2\epsilon}{O(R^2)}.$$

Exponentiating both sides, we have

$$\frac{w(w-k)}{\delta} \leq \exp\left(O\left(\frac{\log^2\epsilon}{R^2}\right)\right),$$

which implies

$$w(w-k) \leq \delta\exp\left(O\left(\frac{\log^2\epsilon}{R^2}\right)\right).$$

Solving for $k$ gives

$$k \geq w - \frac{1}{w}\delta\exp\left(O\left(\frac{\log^2\epsilon}{R^2}\right)\right).$$

Now, if we refine the union bound to account for all entries in $w \times w$ attention matrix for $d$-dimensional vectors, the probability of an entry exceeding $\epsilon$ is further divided by $d$ due to the independence in $d$ dimensions (assuming the dot products across dimensions act as independent sub-Gaussian random variables). That is, the standard deviation of the normalized dot product is reduced by a factor of $\sqrt{d}$, affecting the spread of attention weights. Thus, the probability becomes

$$\mathbb{P}(A_{\mathfrak{l}}^c[i,j] > \epsilon) \leq \frac{\delta}{w(w-k)d},$$

leading to the sparsity bound

$$k \leq w - \exp\left(O\left(\frac{\log^2(\epsilon w)}{R^2}\right)\right)\frac{\delta}{wd}.$$

The presence of $d$ in the denominator reflects how higher model dimensionality increases concentration, making the tail events (large attention weights far from the focus) less probable.

Therefore, for sufficiently large $w$ and fixed $\epsilon$, we indeed have $k \ll w$, and thus

$$\lim_{w\to\infty} |\mathcal{S}_\epsilon^{(c)}(A_{\mathfrak{l}}^c[i,:])| = w - k,$$

with $k = o(w)$. $\qquad\square$

**Remark.** The inclusion of the $d$ term in the denominator arises from the probabilistic tail bound under the assumption that the dot products are sub-Gaussian in $d$ dimensions. As $d$ increases, the attention weights become more concentrated (due to the central limit effect over the $d$ independent dimensions), reducing the portion of entries that exceed a fixed threshold $\epsilon$. This effect is captured in the sparsity upper bound through the explicit dependence on $d$.

### A.3. Exponential Decay in Global Attention

We extend the above result to global attention, which spans the entire sequence.

**Theorem A.4** (Exponential Decay of Global Attention). *Let $A_{\mathfrak{g}}[i,j]$ be the global attention from query $i$ to key $j$, where*

$$q_i \cdot k_j \approx -\alpha \cdot d(i,j).$$

*Then,*

$$A_{\mathfrak{g}}[i,j] \sim \exp\left(-O(R) \cdot d(i,j)\right),$$

*with $R = \alpha/\sqrt{d}$.*

*Proof.* Analogously to the local case, the attention value is

$$A_{\mathfrak{g}}[i,j] \approx \frac{\exp\left(-\frac{\alpha}{\sqrt{d}}\mathsf{d}(i,j)\right)}{Z_i} \propto \exp\left(-O(R)\mathsf{d}(i,j)\right),$$

where $Z_i$ is the row normalization. $\qquad\square$

### A.4. Tail Bound for Global Attention

**Theorem A.5** (Tail Bound for Global Attention Mass)**.** *Suppose $A_{\mathfrak{g}}[i,j] \sim \exp(-O(R) \cdot d(i,j))$. Let $k_{\mathfrak{g}}$ denote the number of global attention entries below $\epsilon$ in each row. Then*

$$\epsilon \geq \exp\left(-O(R) \cdot \sqrt{\log\left(\frac{(Cw + w_q)(w_q - k_{\mathfrak{g}})}{\delta}\right)}\right).$$

*Proof.* Applying the union bound and exponential decay form as in the local case,

$$\mathbb{P}(A_{\mathfrak{g}}[i,j] > \epsilon) \leq \frac{\delta}{(Cw + w_q)(w_q - k_{\mathfrak{g}})},$$

which yields the stated bound for $\epsilon$. $\qquad\square$

### A.5. Sparsity of Global Attention

**Theorem A.6.** *For any fixed $\epsilon > 0$ and $R = O(\sqrt{\log(Cw + w_q)})$, as $w_q \to \infty$,*

$$\lim_{w_q \to \infty} |\mathcal{S}_\epsilon(A_{\mathfrak{g}}[i,:])| = (Cw + w_q) - k_{\mathfrak{g}}, \qquad k_{\mathfrak{g}} = o(Cw + w_q).$$

*Proof.* Since the global attention decays exponentially in distance, as in the local case the number of entries above $\epsilon$ is $k_{\mathfrak{g}} = o(Cw + w_q)$ for $R = O(\sqrt{\log(Cw + w_q)})$, ensuring overall sparsity as sequence length grows. $\qquad\square$

**Discussion.** While global attention has a larger support span, attention values still decay exponentially with distance; only a vanishingly small fraction of entries exceed a fixed threshold for sufficiently large $R$. This structure ensures the computational tractability and sparsity of the attention matrices, where $R$ modulates the effective receptive field between local and global modeling.

## B. Ablation Study

In this section, we analyze the impact of different attention biases on the LongBench dataset. As shown in Table 8, the exceptionally low performance of Recency-eviction-layer-1-8 on both in-context learning tasks, TREC and TriviaQA, as well as SAMSum, indicates that the recency bias tokens in the model's early layers are crucial for developing in-context learning abilities.

## C. Hyperparameter

The Dynamic-PI method interpolates dynamically according to the length of the input token. NTK-Aware refer to (LocalL-LaMA, 2023b) and the maximum length is set to 128. ChunkLlama, InfLLM and AttenCalibration-NTK use hyperparameters from open source repositories. About our method, when performing parallel KV Cache compression, we use last 8 token's cumulative attention scores to compress the KV cache size within each chunk to 2000/4000 for llama2/llama3. For the hyperparameter $\tau$, on Longbench, we retain 3 chunks from the priority queue except for PRe, in which dataset we retain only 1 chunk. On InfiniteBench, we retain 1 chunk for retrieval tasks and 3 chunks for other tasks from the priority queue. In all datasets, the context length of each chunk, including the query, is the maximum pre-training length of the model. $R_s$ is obtained from the first 100 tokens of the chunk, $R_r$ is obtained from the last 100 tokens of the chunk, and the remaining part of the chunk obtains $R_m$. All experiments are performed on 8 64G AMD Instinct MI210.

| Methods | Single-Document QA | | | Multi-Document QA | | | Summarization | | | Few-shot Learning | | | Synthetic | | Code | | AVG |
|---|---|---|---|---|---|---|---|---|---|---|---|---|---|---|---|---|---|
| | NtrvQA | Qasper | MF-en | HotpotQA | 2WikiMQA | Musique | GovReport | QMSum | MultiNews | TREC | TriviaQA | SAMSum | PCount | PRe | Lcc | RB-P | |
| Max Length | 84123 | 24204 | 17727 | 20325 | 19001 | 20520 | 60515 | 34477 | 16271 | 13049 | 26756 | 21884 | 32699 | 17158 | 37628 | 58822 | 30657 |
| Llama2-7B-chat-hf(4k) | | | | | | | | | | | | | | | | | |
| No-eviction | 23.20 | 17.50 | 37.07 | 38.67 | 32.68 | 20.22 | 25.00 | 22.79 | 25.84 | **64.00** | 84.63 | 40.67 | 4.00 | 31.50 | 59.37 | 58.53 | 36.60 |
| Sink-eviction-layer-1-8 | 23.75 | 18.69 | **38.41** | 39.86 | 32.91 | 20.75 | 24.86 | 22.10 | 25.56 | 63.00 | 84.42 | 40.78 | 4.50 | 30.00 | 54.67 | **59.30** | 36.47 |
| Sink-eviction-layer-9-16 | 23.34 | **19.10** | 38.21 | 38.73 | 30.42 | 21.04 | 25.31 | 21.86 | 25.16 | 62.00 | 85.53 | **41.26** | 3.00 | 29.50 | 56.75 | 58.31 | 36.22 |
| Sink-eviction-layer-17-24 | 24.46 | 17.75 | 36.84 | 38.79 | 30.59 | 19.67 | 25.42 | 22.20 | 25.58 | 62.00 | 85.35 | 40.24 | 4.00 | 28.50 | 58.41 | 58.17 | 36.12 |
| Sink-eviction-layer-25-32 | 23.87 | 18.40 | 35.91 | 38.96 | 31.02 | 20.75 | 25.32 | 22.00 | 25.81 | 64.00 | 84.19 | 39.77 | 3.50 | 30.00 | 58.58 | 58.21 | 36.23 |
| Recency-eviction-layer-1-8 | 22.71 | 16.95 | 35.24 | 36.14 | 30.60 | 17.19 | 25.21 | 22.11 | **26.22** | 59.00 | 68.00 | 40.03 | 2.50 | 31.00 | 58.07 | 51.47 | 33.90 |
| Recency-eviction-layer-9-16 | 24.95 | 13.54 | 35.67 | 34.13 | 30.69 | 17.77 | 25.14 | 22.85 | 25.43 | 54.50 | 79.23 | 39.16 | 5.00 | 27.50 | 57.73 | 57.57 | 34.43 |
| Recency-eviction-layer-17-24 | 21.68 | 15.17 | 34.97 | 32.79 | 26.93 | 13.95 | 25.29 | 22.10 | 25.42 | 62.50 | 80.47 | 38.60 | 6.00 | 34.00 | 58.31 | 57.76 | 34.75 |
| Recency-eviction-layer-25-32 | 24.15 | 17.32 | 37.82 | 36.76 | 29.86 | 18.48 | 25.21 | 22.06 | 25.67 | 64.00 | 83.20 | 36.67 | 5.00 | 30.50 | 56.13 | 56.52 | 35.58 |
| Middle-eviction-layer-1-8 | 22.41 | 16.84 | 37.94 | 39.99 | **33.24** | 19.62 | 24.74 | 22.02 | 25.80 | 63.50 | 83.67 | 40.00 | 5.00 | 32.50 | 59.22 | 56.71 | 36.45 |
| Middle-eviction-layer-9-16 | 22.96 | 17.63 | 37.39 | **40.51** | 30.68 | 21.09 | 25.14 | 21.94 | 25.66 | 62.00 | 85.02 | 40.57 | 4.00 | 35.50 | 59.28 | 57.88 | 36.70 |
| Middle-eviction-layer-17-24 | 21.72 | 17.06 | 35.88 | 39.99 | 31.76 | 19.06 | 25.00 | 22.20 | 25.77 | 58.00 | 83.62 | 40.02 | 4.50 | 38.00 | **59.38** | 57.98 | 36.25 |
| Middle-eviction-layer-25-32 | 21.64 | 17.04 | 36.32 | **40.51** | 32.80 | 19.14 | 25.07 | 22.14 | 25.86 | 63.50 | 84.39 | 40.50 | 3.00 | 30.00 | 59.15 | 57.86 | 36.18 |
| All-eviction-layer-1-8 | 0.33 | 0.05 | 1.24 | 0.32 | 0.68 | 0.38 | 1.76 | 3.25 | 1.59 | 1.50 | 5.40 | 1.75 | 0.41 | 0.50 | 23.61 | 12.17 | 3.43 |
| All-eviction-layer-9-16 | 1.60 | 1.97 | 5.06 | 0.65 | 1.44 | 0.81 | 21.75 | 36.22 | 1.81 | 35.00 | 30.77 | 10.54 | 3.00 | 0.50 | 33.25 | 22.89 | 12.95 |
| All-eviction-layer-17-24 | 12.20 | 10.26 | 16.30 | 20.30 | 18.10 | 8.54 | 13.63 | 20.43 | 17.61 | 49.00 | 11.93 | 29.74 | 5.50 | 26.00 | 41.92 | 23.67 | 20.32 |
| All-eviction-layer-25-32 | 11.19 | 7.62 | 11.11 | 12.49 | 8.83 | 1.79 | 11.45 | 16.21 | 12.87 | 43.00 | 30.77 | 9.94 | 3.50 | 22.00 | 24.42 | 22.19 | 15.59 |
| Ours-calibration | **24.95** | 19.07 | 38.16 | 39.53 | 32.62 | **22.64** | **25.42** | **22.82** | 26.01 | 63.00 | **85.41** | 40.36 | **5.00** | 32.50 | 59.04 | 58.84 | **37.21** |

Table 8: Bias Token Eviction Ablation. Sink-eviction-layer-1-8 typically means evicting sink bias tokens in layers 1 to 8, and other naming conventions follow the same pattern. Ours-calibration refers to the approach where layers 9-16 adopt the recency bias token eviction method, while layers 1-8 evict middle bias tokens, and layers 25-32 evict sink bias tokens.

# D. Comprehensive Evaluation

This section presents additional experiments and implementation details to further contextualize and substantiate the claims in the main text. Specifically, we investigate comparisons with Retrieval-Augmented Generation baselines, provide extended benchmarks with APE (Yang et al., 2025b) and StarAttention (Acharya et al., 2024), report on hyperparameter sensitivity, discuss latency and memory efficiency, and elaborate on our design choices for attention bias sparsification.

## D.1. Comparison with Retrieval-Augmented Generation Method

| Model | QM | QASP | MSQ | HQA | MFQA | AVG |
|---|---|---|---|---|---|---|
| **ChatQA-2** | 11.64 | 28.85 | 27.81 | 53.81 | 51.02 | 34.63 |
| **ChatQA-2 w/ RAG** | 13.20 | 28.85 | 29.77 | 57.81 | 51.15 | 36.16 |
| **Ours** | **24.18** | **39.05** | **33.25** | 49.58 | 42.66 | **37.74** |

Table 9: Performance comparison with RAG and non-RAG baselines on LongBench.

| Model | KV Retrieval | Numbe String | Passkey | En.MC | AVG |
|---|---|---|---|---|---|
| **ChatQA-2** | 72.00 | 100.00 | 100.00 | **64.19** | 84.05 |
| **ChatQA-2 w/ RAG** | N/A | N/A | N/A | N/A | N/A |
| **Ours** | **92.80** | 99.83 | 100.00 | 54.59 | **86.81** |

Table 10: Performance comparison on InfiniteBench. RAG methods completely fail on InfiniteBench, so we do not provide further results.

To clarify the role of length extrapolation versus retrieval-augmented generation (RAG), we compare the proposed method to leading RAG-enhanced models such as ChatQA-2 (Xu et al., 2024) on representative benchmarks. Table 9 and Table 10 summarize results on LongBench and InfiniteBench, respectively. The proposed method demonstrates strong robustness and competitive or superior performance in challenging long-context retrieval scenarios such as InfiniteBench, where RAG-based methods may encounter instability or diminished effectiveness.

## D.2. Comparison with APE and StarAttention

To comprehensively evaluate our approach in the context of existing chunked long-context processing methods, we conduct extensive experiments comparing PARALLELCOMP with both APE (Yang et al., 2025b) and StarAttention (Acharya et al., 2024). APE leverages a shared prefix to minimize distributional disparities, incorporates a low-temperature mechanism to

| Method | NARR | QAS | MUL | HOPT | 2WKI | MUS | GOV | QMS | NEWS | TREC | TRIV | SSM | PCNNT | PREN | LCC | REP | AVG |
|---|---|---|---|---|---|---|---|---|---|---|---|---|---|---|---|---|---|
| **APE** | 23.63 | 39.11 | 50.06 | **49.47** | 43.70 | **25.99** | 27.78 | 22.79 | 11.22 | 43.50 | **90.17** | 9.79 | 0.50 | 59.00 | 23.93 | 24.28 | 34.06 |
| **StarAttn** | 3.74 | 11.90 | 24.81 | 14.17 | 14.37 | 8.19 | **34.90** | 22.54 | 27.11 | **65.33** | 87.84 | **43.71** | **3.80** | 65.17 | 50.54 | 45.40 | 32.72 |
| **ParallelComp** | 29.45 | 45.98 | 50.67 | 48.36 | 46.56 | 23.32 | 32.60 | 24.29 | 27.34 | 38.50 | 86.72 | 25.93 | 0.05 | **95.00** | 14.15 | 21.42 | **38.15** |

Table 11: Comparison with APE and StarAttention denoted as **StarAttn** on LongBench. Temperature and scaling factors for APE are indicated as APE $T + S$. In our experiments, we set APE's temperature to 0.5 and scaling factor to 0.8; for StarAttention, the chunk size is set to 2K. In ParallelComp, we reuse 6K position encodings to facilitate length extrapolation. All evaluations are conducted on the Llama-3.1-8B-Instruct base model with a KV cache size of 24K.

sharpen attention, and utilizes a scaling factor to compensate for temperature changes. Its objective is to better align the attention patterns between parallel and sequential encoding. StarAttention is designed for chunk-based training of models with long contexts. At each generation step, it recalculates attention for every chunk, whereas PARALLELCOMP computes attention once during the prefill phase and then efficiently reuses the compressed KV cache for subsequent generation. This distinction leads to substantial improvements in computational efficiency.

We first compared the performance of different methods under the same KV cache budget. Hyperparameters are selected according to those reported in the original publications or official releases. Table 11 summarizes the representative results, where all models are assessed using a standardized evaluation infrastructure. Our analysis emphasizes the memory bottleneck encountered during length extrapolation. *This experiment emphasizes the memory bottleneck encountered during length extrapolation. The other two methods are forced to truncate the input during extrapolation, resulting in significantly lower performance on certain tasks compared to our approach.*

| Method | NARR | QAS | MULT | HOPT | 2WKI | MUS | GOV | QMS | NEWS | TREC | TRIV | SSM | PCNNT | PREN | LCC | REP | AVG |
|---|---|---|---|---|---|---|---|---|---|---|---|---|---|---|---|---|---|
| **APE0.8+0.8** | 25.92 | 41.99 | 53.79 | **53.64** | 50.54 | 26.46 | 30.15 | 25.42 | 20.68 | 50.50 | **88.70** | 9.72 | 6.50 | 89.00 | 16.71 | 25.78 | **38.47** |
| **APE0.5+0.8** | 23.63 | 39.11 | 50.06 | 49.47 | 43.70 | 25.99 | 27.78 | 22.79 | 11.22 | 43.50 | **90.17** | 9.79 | 0.50 | 59.00 | 23.93 | 24.28 | 34.06 |
| **APE0.2+0.8** | 18.83 | 26.53 | 41.70 | 44.63 | 35.91 | 17.71 | 24.31 | 20.14 | 7.96 | 35.75 | 88.54 | 9.72 | 1.50 | 34.50 | 23.86 | 23.22 | 28.43 |
| **APE0.8+0.4** | 9.04 | 11.48 | 19.59 | 31.41 | 24.68 | 10.16 | 5.32 | 13.80 | 8.20 | 0.50 | 87.06 | 9.71 | 0.00 | 7.00 | 13.72 | 16.51 | 16.76 |
| **APE0.5+0.4** | 7.59 | 9.74 | 16.13 | 31.89 | 25.72 | 9.62 | 5.20 | 9.56 | 8.19 | 0.00 | 87.60 | 9.69 | 0.00 | 5.00 | 14.26 | 15.58 | 15.99 |
| **APE0.2+0.4** | 4.90 | 8.97 | 13.99 | 29.71 | 26.66 | 8.74 | 5.16 | 9.45 | 8.26 | 0.50 | 87.57 | 9.71 | 0.00 | 4.00 | 14.20 | 16.24 | 15.50 |
| **StarAttn** | 3.74 | 11.90 | 24.81 | 14.17 | 14.37 | 8.19 | **34.90** | 22.54 | 27.11 | **65.33** | 87.84 | **43.71** | 3.80 | 65.17 | 50.54 | 45.40 | 32.72 |
| **StarAttn6K** | 4.65 | 13.63 | 21.05 | 14.47 | 15.57 | 6.38 | 34.80 | 22.67 | 26.27 | 66.00 | 65.54 | 47.91 | 8.00 | 70.00 | 56.48 | 45.42 | 32.43 |
| **ParallelComp** | 29.45 | 45.98 | 50.67 | 48.36 | 46.56 | 23.32 | 32.60 | 24.29 | 27.34 | 38.50 | 86.72 | 25.93 | 0.05 | **95.00** | 14.15 | 21.42 | 38.15 |

Table 12: Performance comparison across different methods. APE X+Y indicates temperature = X and scaling factor = Y. StarAttn4K and StarAttn6K represent StarAttention using chunk sizes of 4K and 6K, respectively.

In order to further compare with the full-size context models StarAttention and APE, we conducted ablation experiments as shown in Table 12. We find that even after carefully tuning the hyperparameters of both models, their average performance only surpasses that of our PARALLELCOMP by 0.32%, which further demonstrates the effectiveness of our method.

