# OpenReview forum: "ParallelComp: Parallel Long-Context Compressor for Length Extrapolation"
_ICML.cc/2025/Conference — ICML 2025 poster_

### Official Review · Reviewer_P8gm · 2025-03-03

**Overall Recommendation:** 3

**Summary:**

ParallelComp processes the input sequence chunk by chunk independently. Therefore, they can parallel process the attention matrix, except for the last chunk of attention.

**Claims And Evidence:**

No, the supporting evidence is not clear and not well connected to their claims.

**Essential References Not Discussed:**

StarAtteniton (https://arxiv.org/pdf/2411.17116) and APE (https://arxiv.org/pdf/2502.05431) look extremely similar to this work. I strongly suggest that the authors compare ParallelComp with those baselines.

**Experimental Designs Or Analyses:**

Using Llama 2 looks like a bad choice in 2025. At least, I think we should use Gemma 2 or EXAONE 3.5 for short context models.

I wonder how performance is good in long context language models such as llama 3.1. Since the KV cache memory is usually a more significant problem rather than latency in most scenarios (up to 1M token context), we can compare this method to long context LLMs that use the same amount of KV cache memory with ParallelComp.

I wonder how performance is good when we apply ParallelComp to long context LLM such as Llama 3.1 and Qwen 2.5. Llama 2 and 3 are pretty old and short at this point.

**Methods And Evaluation Criteria:**

No, I think the evaluation of the calibration and compression variant is not well justified because the performance looks pretty similar between whole variants in Llama 3.

**Other Comments Or Suggestions:**

In the abstract, I think we need to avoid the bold text if it is unnecessary.

Tables 1, 2, 3, 4, and 5 are too tiny text. Please revise the overall formatting significantly.

Promoting performance between GPT4 might be overclaiming. The promotion that ParallelComp can make 8B better than GPT4 may lead to misunderstanding that this method improves the long context performance. I feel afraid of such overclaims, so I gently ask you to think about this once more.

In Tables 1 and  2, I think we should show the average length of the dataset or 99% of the length of the dataset because the maximum length might show only the outliers.

In Figure 2, the terms `Key` and `Value` should be plural (-s).

**Other Strengths And Weaknesses:**

Please refer to other sections.

**Questions For Authors:**

Please refer to the upper sections.

**Relation To Broader Scientific Literature:**

I am not sure the key contribution of this work (chunkwisly parallel encoding) is scientifically impactful.

I am not sure if Parallel Attention Bias is effective in terms of downstream tasks.

**Theoretical Claims:**

Yes.

However, I am concerned that Parallel Attention Bias is necessary to sparsify attention mechanisms.

---

> ### Author Rebuttal · Authors · 2025-04-01
>
> Thank for your suggestions!
>
> Q1：**Key differences**
>
> |Method|NARR|QAS|MUlT|HOPT|2WKI|MUS|GOV|QMS|NEWS|TREC|TRIV|SSM|PCNNT|PREN|LCC|REP|AVG|
> |------|----|---|----|----|----|---|---|---|----|----|----|---|-----|----|---|---|----|
> |APE|23.63|39.11|50.06|49.47|43.70|25.99|27.78|22.79|11.22|43.50|90.17|9.79|0.50|59.00|23.93|24.28|34.06|
> |Star|3.74|11.90|24.81|14.17|14.37|8.19|34.90|22.54|27.11|65.33|87.84|43.71|3.80|65.17|50.54|45.40|32.72|
> |ParallelComp|29.45|45.98|50.67|48.36|46.56|23.32|32.60|24.29|27.34|38.50|86.72|25.93|0.05|95.00|14.15|21.42|38.15|
> **Table 1:** Comparison with other methods.
>
> - We focus on the **memory-bound** encountered during the implementation of length extrapolation.
> - ParallelComp introduces a **two-stage compression strategy**: chunk-level kv cache eviction followed by intra-chunk compression. **Attention calibration** is applied to reduce the performance loss caused by compression. This enables both context window extension and improved inference throughput.
> - **APE** consists of a shared prefix to reduce distributional differences, a low-temperature mechanism to sharpen attention, and a scaling factor to adjust for temperature-induced changes. It aims to align the attention patterns of parallel and sequential encoding.
> - **StarAttention** targets models with native long-context support. It recalculates attention for each chunk at every step, while our method computes attention only once at the prefill stage and reuses the **compressed**  KV cache for generation—greatly improving efficiency.
>
>
>
> - We conducted a comparative experiment, with the code implementation based on [1,2]. However, APE lacks prompt design, Star is missing the Longbench experiment, and hyperparameter configurations are absent in both. For APE, we set the temperature to 0.5 and the scaling factor to 0.8. For Star, the chunk size was set to 2K. In our approach, we reuse these 6K position encodings for length extrapolation. Base model is Llama-3.1-8B-Instruct.
>
>
>
> Q2: **Supplementary baseline**
>
>
>
> |Method|NARR|QAS|MUlT|HOPT|2WKI|MUS|GOV|QMS|NEWS|TREC|TRIV|SSM|PCNNT|PREN|LCC|REP|AVG|
> |------|----|---|----|----|----|---|---|---|----|----|----|---|-----|----|---|---|----|
> |Llama3.1|27.80|44.25|49.46|47.86|40.54|23.64|32.64|22.90|26.90|38.00|88.44|25.64|2.02|92.00|10.35|18.64|36.94|
> |ParallelComp-Llama3.1|29.45|45.98|50.67|48.36|46.56|23.32|32.60|24.29|27.34|38.50|86.72|25.93|0.05|95.00|14.15|21.42|38.15|
> |Qwen2.5|27.83|41.31|50.41|53.52|44.68|30.00|33.38|24.01|25.40|71.00|86.10|39.91|7.25|100.00|6.86|7.88|40.60|
> |ParallelComp-Qwen2.5|28.42|42.24|50.54|56.26|42.02|28.25|33.43|23.20|25.20|71.50|89.21|41.84|5.00|93.50|20.73|13.34|41.54|
> **Table 2:** Longbench.
>
> |Method|PS|NUM|KV|EN.MC|MATH|CODE|AVG|
> |------|---|---|--|-----|-----|-----|----|
> |Llama3.1|5.59|26.25|18.60|32.86|31.52|22.56|26.36|
> |ParallelComp-Llama3.1|100.00|83.56|88.60|66.38|37.14|22.08|59.55|
> |Qwen2.5|59.32|58.31|33.80|61.39|85.71|23.76|53.72|
> |ParallelComp-Qwen2.5|100.00|76.27|63.40|66.86|92.57|24.75|70.64|
> **Table 3:** Infinitebench.
>
> - The KV cache management mechanism of Gemma2 is significantly different from that of LLaMA and Qwen models, so we have not yet fully implemented it. If we have more time, we promise to submit the test results of EXAONE 3.5 and Gemma2. We promise to provide it in a future version of the paper.
>
>
> - We conduct experiments on Llama3.1 and Qwen2.5 using Longbench and InfiniteBench under the same 24K KV cache budget. While other models use 24K position encodings directly, our approach leverages extrapolation techniques to reuse 6K position encoding. Others use the original length.In the experiment, it is found that whether the prompt word(such as “<|begin_of_text|>” for Llama3.1) is used has a great influence on the performance, and we make a comparison in the case of the prompt word.
> It can be seen that, under the same 24K KV cache size budget, our approach still outperforms other models that **do not implement extrapolation**, especially on the InfiniteBench dataset.
>
>
>
> Q3: **Necessity to sparsify attention mechanisms**
>
> Based on our observations, certain tokens at specific layers do indeed contribute to improving the model's performance:
>
> - In early layers, attention biases are crucial for context understanding—removing them harms performance.
> - For code tasks, biases in middle layers matter most—removal also leads to performance drops.
> - The attention sink boosts retrieval in early layers but hurts performance in later layers.
>
> However, in other layers, **these attention biases significantly hinder the model's ability to process long contexts**. Therefore, removing these anomalous tokens can improve the model's performance.
>
> Q4: **Others**
>
> - We will consider your suggestions and incorporate additional supplementary materials into future versions of the paper, especially the release of the open-source code.
>
>
>
> **Reference**:
>
> [1]https://github.com/Infini-AI-Lab/APE
>
> [2]https://github.com/NVIDIA/Star-Attention

---

> > ### Comment · Reviewer_P8gm · 2025-04-01
> >
> > Thank you for your rebuttal.
> >
> > I understand more about this method, compared to APE and Start Attention, by the authors' explanations.
> >
> > However, I still have concerns about performance evaluation compared to existing parallel encoding works. What is the model used in Table 1? It seems the reported numbers are pretty different from the APE paper (Table 7). What is the chunk size of APE? Why use different chunk sizes between StartAttention and yours? Is it because of latency? And in comparison, I think we are missing a report on the latency of methods. How fast or slow your method compared to FlashAttention, InfLLM, APE, and StarAttention?
> >
> > I am looking forward to the authors' response. Thank you again for your great work.

---

> > > ### Author Response · Authors · 2025-04-03
> > >
> > > Thank you for your feedback and comments! We commit to citing the APE and StartAttention in future version!
> > >
> > > Q1: **Hyperparameter Settings**
> > >
> > > |Method|NARR|QAS|MULT|HOPT|2WKI|MUS|GOV|QMS|NEWS|TREC|TRIV|SSM|PCNNT|PREN|LCC|REP|AVG|
> > > |------|----|----|-----|-----|-----|-----|-----|-----|-----|-----|-----|-----|------|-----|-----|-----|-----|
> > > |APE0.8+0.8|25.92|41.99|**53.79**|**53.64**|**50.54**|**26.46**|30.15|**25.42**|20.68|50.50|88.70|9.72|6.50|89.00|16.71|25.78|**38.47**|
> > > |APE0.5+0.8|23.63|39.11|50.06|49.47|43.70|25.99|27.78|22.79|11.22|43.50|**90.17**|9.79|0.50|59.00|23.93|24.28|34.06|
> > > |APE0.2+0.8|18.83|26.53|41.70|44.63|35.91|17.71|24.31|20.14|7.96|35.75|88.54|9.72|1.50|34.50|23.86|23.22|28.43|
> > > |APE0.8+0.4|9.04|11.48|19.59|31.41|24.68|10.16|5.32|13.80|8.20|0.50|87.06|9.71|0.00|7.00|13.72|16.51|16.76|
> > > |APE0.5+0.4|7.59|9.74|16.13|31.89|25.72|9.62|5.20|9.56|8.19|0.00|87.60|9.69|0.00|5.00|14.26|15.58|15.99|
> > > |APE0.2+0.4|4.90|8.97|13.99|29.71|26.66|8.74|5.16|9.45|8.26|0.50|87.57|9.71|0.00|4.00|14.20|16.24|15.50|
> > > |StarAttention4K|3.74|11.90|24.81|14.17|14.37|8.19|**34.90**|22.54|27.11|65.33|87.84|43.71|3.80|65.17|50.54|45.40|32.72|
> > > |StarAttention6K|4.65|13.63|21.05|14.47|15.57|6.38|34.80|22.67|26.27|**66.00**|65.54|**47.91**|**8.00**|70.00|**56.48**|**45.42**|32.43|
> > > |Ours|**29.45**|**45.98**|50.67|48.36|46.56|23.32|32.60|24.29|**27.34**|38.50|86.72|25.93|0.05|**95.00**|14.15|21.42|38.15|
> > >
> > > **Table 1.** `APE X+Y` indicates temperature = X and scaling factor = Y.
> > >
> > > **APE**
> > > - We followed the original papers as closely as possible and ran multiple experiments with different settings.
> > > - We set the chunk size as **6000**.
> > > - **Temperature and Scaling**:  A temperature of `0.8` and a scaling factor of `0.8` yielded results closest to the original paper and slightly better than our method under the same unified settings.
> > > - **KV Cache**: It’s worth noting that **APE does not implement any KV cache eviction strategy**, so it uses a full size KV cache, which gives it a performance edge.
> > >
> > > **StarAttention**
> > > - We use a chunk size of `4K` tokens as reported in the original paper.
> > > - To make a fair comparison, we also tried a `6K` chunk size, which reduced its performance.
> > >
> > > **Summary**
> > > - APE achieved the best performance among all baselines with the APE0.8+0.8 hyperparameter setting. Although our method performs slightly worse, considering that we only used the compressed KV cache for inference, these performance losses are acceptable.
> > > - The reason why StarAttention is particularly sensitive to chunk size is unknown.
> > > - The APE method appears sensitive to the scaling factor hyperparameter, and the reason is unclear. **I look forward to further discussing this issue with you.**
> > >
> > > Q2:  **Parallel Design Implementation Details**
> > >
> > > - All windows are padded to the same length for parallel processing, followed by a forward pass to compute the KV cache, with optional in-chunk compression. The compressed KV caches are then added to a priority queue for eviction based on their self-information.
> > >
> > >
> > > - In contrast to APE and StarAttention, our chunk and token KV cache eviction strategy effectively mitigates out-of-memory (OOM) issues, making our approach more scalable and memory-efficient, especially for long-context scenarios.
> > >
> > >
> > > Q3: **Evaluation on NarrativeQA**
> > >
> > >
> > >
> > > |Method|Effective Number Of Samples|OOM Samples|Inference Time Per Sample (s)|Accuracy|
> > > |------|------------------------|----------|------------------------|----------|
> > > |APE-truncation|85|115|6.06|25.92|
> > > |StarAttention|200|0|10.28|4.65|
> > > |InfLLM|200|0|12.28|26.58|
> > > |Ours|200|0|1.84|29.45|
> > > |Ours-truncation|200|0|1.76|27.60|
> > > |Ours-compression|200|0|1.03|28.36|
> > >
> > > **Table 2.** Latency and performance comparison. For the OOM samples, we performed a truncation operation on APE. Truncation involves retaining the first and last halves of a sample after reducing it to the target length(36k).
> > >
> > > | Method| Prefill(s) | Tokens/s | Generate(s) | Tokens/s |
> > > |--------------|------|------|-------|----------|
> > > | FullKV|1206.55| 4937.86 |260.68      | 140.69   |
> > > | Ours|123.65 | 48182.47  |245.20   |  152.57   |
> > > | Ours-Comp| 62.32| 95599.52  | 143.59      | 265.50   |
> > > **Table 3.** Inference latency and Throughput.
> > >
> > >
> > > All experiments are conducted on an AMD Instinct MI210 64GB GPU. FullKV means using full-attention reasoning experiments. Max length is set to 36k. “Ours” only uses chunk eviction and “Ours-compression” uses chunk eviction and parallel KV cache eviction. We evaluate performance using the following metrics:
> > >   - **Effective Number of Samples**: Successfully processed samples.
> > >   - **OOM Samples**: Failed samples due to memory issues.
> > >   - **Inference Time per Sample**: Average processing time per instance.
> > >   -   KV cache compression **accelerated prefill time** and **improve throughput**.
> > >
> > > Q4: **Note on FlashAttention**
> > >
> > > -  **FlashAttention** is orthogonal to our method, which focuses on **infra improvements** such as memory management.

---

### Official Review · Reviewer_GVLE · 2025-03-12

**Overall Recommendation:** 3

**Summary:**

This paper introduces ParallelComp, which the authors claim to be able to extend the context window of off-the-shelf LLMs from 4K to 128K tokens while maintaining computational efficiency. The authors address the critical challenge of attention sink phenomena in parallel attention mechanisms, where models disproportionately focus on initial/recent tokens, impairing long-context understanding.


## Update After Rebuttal

I have read the authors' rebuttal and decided to keep the current positive score.

**Claims And Evidence:**

1. Parallel Compression Framework. The authors propose a chunk-based architecture with parallel local attention computations and strategic KV cache eviction. This enables efficient processing of ultra-long contexts on a single A100 GPU.

2. Attention Bias Analysis & Calibration. The authors have conduct a systematic identification of three attention bias patterns in parallel mechanisms.

**Essential References Not Discussed:**

As I am not well-versed in this area, I have not identified any critical missing references.

**Experimental Designs Or Analyses:**

1. (Apologies if this question is somewhat naïve, as I am extremely unfamiliar with this field.) The authors conduct a thorough comparison against existing length extrapolation methods for tasks such as single/multi-document QA. However, these tasks could potentially benefit from RAG without requiring long-context extrapolation in LLMs. Would it be possible to compare the proposed approach with leading RAG methods such as [1,2]?


2. Does the paper provide a detailed description of the calibration data used? Have they analyzed stability across different context sources (e.g., code vs. narrative text)?


---
[1] RAPTOR: recursive abstractive processing for tree-organized retrieval.

[2] ChatQA 2: Bridging the Gap to Proprietary LLMs in Long Context and RAG Capabilities

**Methods And Evaluation Criteria:**

Regarding the methods, the proposed methods align well with the core challenge of long-context extrapolation. The parallel attention mechanism and chunk eviction strategies are designed to address the memory/computation bottlenecks of long sequences. In the meanwhile, compatibility with flash attention is well attended to.

The evalution seems comprehensive and appropriate for me. I am just wondering whether the authors have considered testing their method on more cross-lingual datasets?

**Other Comments Or Suggestions:**

N/A

**Other Strengths And Weaknesses:**

See above

**Questions For Authors:**

N/A

**Relation To Broader Scientific Literature:**

This work focuses on LLM length extrapolation, an important research direction with numerous influential studies. Given that the proposed approach is training-free, its contribution appears particularly meaningful.

**Theoretical Claims:**

The overall theoretical framework provides useful explanation about parallel attention collapse.

---

> ### Author Rebuttal · Authors · 2025-03-31
>
> We would like to express our sincere gratitude for your fruitful suggestions, and for all your questions below!
>
> Q1: **The authors conduct a thorough comparison against existing length extrapolation methods for tasks such as single/multi-document QA. However, these tasks could potentially benefit from RAG without requiring long-context extrapolation in LLMs. Would it be possible to compare the proposed approach with leading RAG methods such as [1,2]?**
>
>
> | Model|QM|QASP|MSQ|HQA|MFQA|Average|
> |-----|------|------|------|------|------|---------|
> | Llama3-ChatQA-2-8B | 11.64| 28.85| 27.81| 53.81| 51.02| 34.63   |
> | w/ RAG|13.20|28.85|29.77|57.81|51.15|36.16|
> | ours-llama3|24.18| 39.05|33.25|49.58|42.66|37.74|
> **Table 1: Experiments on Longbench**
>
>
> | Model| kv_retrieval | numbe_string | passkey | En.MC | Average |
> |-----|--------------|---------------|---------|-------|---------|
> | ours-llama3|92.80| 99.83|100.00|54.59|86.81|
> | Llama3-ChatQA-2-8B|72.00|100.00|100.00| 64.19|84.05|
> **Table 2 Experiments on InfiniteBench.**
>
> - Since Llama3-ChatQA-2-8B produces "Not Available" results on InfiniteBench when using RAG (w/ RAG), we only compared the results of Llama3-ChatQA-2-8B in Table 2.
> - The above shows the comparison results between our method and the representative RAG method, ChatQA 2 [2]. It is clear that our method performs well compared to the strong RAG baseline, especially on the Longbench dataset.
> - The performance collapse of the RAG method on InfiniteBench is an unknown phenomenon, which also demonstrates the robustness of our method.
>
>
>
>
>
>
> Q2: **Does the paper provide a detailed description of the calibration data used? Have they analyzed stability across different context sources (e.g., code vs. narrative text)?**
> - We calibrated the attention scores based on Equation 7, using an online method to evict  anomalous tokens, so there is no need to sample data in advance for calibration.
> - We found that in code tasks, the attention distribution of the intermediate layers are more sensitive to calibration, while in text tasks, such as in-context learning, the attention distribution of the earlier layers are more sensitive to calibration.
>
>
> **Reference**:
> [1] RAPTOR: recursive abstractive processing for tree-organized retrieval.
> [2] ChatQA 2: Bridging the Gap to Proprietary LLMs in Long Context and RAG Capabilities.

---

### Official Review · Reviewer_KipP · 2025-03-12

**Overall Recommendation:** 3

**Summary:**

This paper proposes a method for training-free length extrapolation of LLM i.e. extending an LLM to process sequence longer than the sequence length it is pretrained on. The key idea is to split the input into chunks that fit in the LLM's context window and perform global attention over the chunks. While previous work (e.g. InfLLM) has also explored similar idea, this paper further proposes to mitigate "attention bias" by removing tokens within the chunk that receive (1) high attention scores (sink tokens) and (2) low self-information scores.

**Claims And Evidence:**

Experiments are conducted on multiple long-context datasets (LongBench, Infinite-bench and perplexity of NarrativeQA, showing that the proposed method outperform previously proposed methods (including methods that manipulate position encodings and similar chunk encoding methods like InfLLM).

However, I find the evidence that the proposed calibration and compression methods is helpful to be relatively weak. For all the experiments (table 1, table 2). If I understand correctly, "ours" is the setting where all tokens are encoded in the parallel fashion, and "ours-compression" and "ours-calibration" correspond to setting where tokens are evicted. And in both tables the difference between "ours" and the other two methods are minimal (except for Llama-2-7B-chat-hf(4k) on Infinite-bench).

Further, the difference between "ours" and InfLLM since to be whether all the chunks are included in the global attention, or only some of the chunks are considered (InfLLM only uses subset of chunks to do global attention), which is not surprising as leveraging more information is more accurate.

**Essential References Not Discussed:**

N/A

**Experimental Designs Or Analyses:**

The experiment settings make sense.

**Methods And Evaluation Criteria:**

The proposed benchmark datasets and baselines are suitable for the method.

**Other Comments Or Suggestions:**

* I think it will be helpful to analyze the setting where the proposed calibration and compression method helps compared to "Ours". For instance, it seems like for `En.MC` and `R.KV`, "ours-calibration" improve upon "ours" more compared to the other tasks.

Typo:
* line 71 in Related work section "reecent" ==> "recent"
* equation 6 -- if $R_{l}$ represents the indices corresponding to tokens to evict, doesn't $K_{x}[R_{l}]$ represent the key states to be evicted?

* The text in Table 5 is too small and uneasy to read.

**Other Strengths And Weaknesses:**

N/A

**Questions For Authors:**

* While the attention analysis in Section 4 is nice, it is unclear to me what's the connection of the patterns observed in figure 3 to the proposed methods aside from the idea that tokens with high attention scores are removed?
* For Llama-3-8B-Instruct (8k) performance on LongBench, it seems like some of the numbers for InfLLM and FullKV are different for the numbers reported in the [InfLLM paper](https://arxiv.org/pdf/2402.04617) Table 5 (e.g. NarrativeQA and MF-en), is the setting different causing the discrepancy? The results for InfiniteBench seems to align with that in InfLLM's Table 1.

**Relation To Broader Scientific Literature:**

This work contributes to the line of work which aims to enable language models to process text longer than their pre-training context window. While there are more language models nowadays that is trained to process very long context already, I think the direction is still an important.

**Theoretical Claims:**

I checked the theoretical claims on attention bias for parallel attention computation.

---

> ### Author Rebuttal · Authors · 2025-03-31
>
> We would like to express our sincere gratitude for your fruitful suggestions, and for all your questions below!
>
> Q1: **Minimal Performance Difference Between "Ours" and Other Methods in Token Eviction Settings.**
> - "Ours" implements the eviction scheme for the entire chunk's KV cache.
> - "Ours-compression" refers to further compressing the KV cache within the chunk.
> - "Ours-calibration-compression" attempts to recover performance further through calibration.
>
> "Ours" and "Ours-compression" refer to the methods of adding chunk eviction and parallel KV cache eviction, respectively. Generally, parallel KV cache eviction tends to degrade the performance of many models, so we hope to recover the performance lost during compression by using an attention distribution calibration mechanism, namely "Ours-calibration-compression."
>
>
>
>
> Q2: **Difference Between "Ours" and InfLLM: Global Attention with Full vs. Subset of Chunks.**
> We need to clarify your misunderstanding of our method.
>
> - "Ours" only **applies attention to chunks in the priority queue**, unlike InfLLM, which retains all chunks in the CPU and causes high I/O overhead and slow prefill speeds by swapping the KV cache between the CPU and GPU. In fact, it can attend to all global chunks simultaneously.
>
>
> - InfLLM addresses the memory-bound issue during inference by retaining only the chunk representations, while "Ours" uses a chunk eviction strategy and parallel compresses the KV cache to increase chunk process throughput. **We mainly focus on the KV cache compression strategy in parallel chunk processing**
>
>
>
>
>
> Q3: **Analyzing the Benefits of Calibration and Compression vs. "Ours".**
>
> We will start our analysis from a core motivation, which is the memory-bound issue in the parallel encoding and the attention bias issue aggravated by compression..
> - “Ours“
> Due to GPU memory limits, it's not feasible to store KV caches for many chunks at once. Offloading chunks to the CPU during parallel processing also introduces high I/O latency, making it impractical.  We propose a more general approach based on two principles:
> **Simplicity:** We use negative log-likelihood (NLL) to model the relevance between queries and chunks without designing extra representations. This reduces communication overhead—especially important in multi-GPU setups where I/O is the main bottleneck.
> **Efficiency:** Sorting in the priority queue only involves scalar comparisons with O(log n) time complexity, unlike heavier methods like InfLLM that use O(n²) retrieval operations.
>
> - “Ours-compression“
> Parallel KV cache eviction helps address memory limitations in parallel computation, but our analysis shows it worsens attention bias. To mitigate this, we introduced a simple attention calibration mechanism “**ours-compression-calibration**” for near-lossless compression, making our design effective.
>
>
>
> Q4: **Typo of equation 6.**
>
> - In Eq. 6, [·] represents the eviction operation, which means removing the indices in [·] from K_x.
>
>
> Q5: **Clarifying the Link Between Figure 3 Patterns and the proposed method.**
>
> - Observation1: Based on our observations, due to using almost the entire window length for parallel encoding, the model always exhibits a strong recency bias in the query chunk region.
>
> - Design1 : It indicates that the model relies on the tokens in the query chunk to make next token predictions, which inspired us to use the negative log-likelihood of the tokens in this region to measure the importance of the chunk.
>
>
> - Observation2: Another phenomenon is that the KV cache compression operation **exacerbates attention bias as shown in Figure 3**, as shown in the first image of Figure 5.
>
> - Design2 : Therefore, designing a calibration mechanism is necessary to mitigate the performance loss caused by exacerbated attention bias.
>
>
>
> Q6: **The performance discrepancy for Llama-3-8B-Instruct (8k) on LongBench may be due to different settings, while results for InfiniteBench align with InfLLM's Table 1.**
>
> - LongBench: For LongBench, we do not use the original hyperparameters reported in the InfLLM paper. Instead, we re-evaluated all baselines under a unified testing framework to ensure fair comparison. Specifically, we adopted the evaluation setup from the open-source repo[1], and ported that framework to the InfLLM codebase for consistency. This was done to eliminate inconsistencies in evaluation scripts, precision settings, and other hyperparameters.
> The inference details
> temperature=0, and num_beams=1, float16 precision.
> - For InfLLM, we use the default hyperparameters as provided in their official code repository to evaluate on longbench.
>
>
>
>
> **Reference**:
> [1] https://github.com/Zefan-Cai/KVCache-Factory.

---

### Official Review · Reviewer_JBTb · 2025-03-13

**Overall Recommendation:** 3

**Summary:**

Motivation is also for better generalization on long seqs. And to use existing LLMs and extend their attention context size. And for more efficient attention on long seqs.

The proposed method does not need any finetuning. Any existing LLM can be used, with the attention mechanism and caching adapted.

The proposed method is called ParallelComp. It uses chunk-based attention for fast parallel calculation to be able to filter out (evict) non-relevant parts of the cache/history in an efficient parallel way, and only then computes global attention on the remaining cache.

Many experiments are performed to show how well it performs on length extrapolation, and results look good for ParallelComp.

## Update after rebuttal

While it is clarified that the code is being released, which is am important point, overall I keep my current score, as I'm not sure about the importance/impactfulness of the presented approach, as also pointed out by other reviewers, and the experiments could also be extended to some more recent LLMs.

**Claims And Evidence:**

Better length extrapolation is claimed by the new proposed method. This is tested and verified in the experiments.

**Essential References Not Discussed:**

-

**Experimental Designs Or Analyses:**

Tested on benchmarks which test for length extrapolation, so they make sense.

**Methods And Evaluation Criteria:**

Tested on benchmarks which test for length extrapolation, so they make sense.

**Other Comments Or Suggestions:**

Sec 2 "reecent" typo.

You have a "bloc97" reference. That is the GitHub username of Bowen Peng (see for example: https://arxiv.org/pdf/2411.19870).

The appendix has the title "Submission and Formatting Instructions for ICML 2025".

**Other Strengths And Weaknesses:**

Strengths:

- Presented approach should be simple to do. Does not need any finetuning.
- Presented approach gives good results.
- Many experiments are performed.

Weaknesses:

- No code released to reproduce the experiments?
- Some parts are a bit unclear to me. See questions below.

**Questions For Authors:**

Where is the code of the experiments? Will this be published?


Fig 1, is that one selected attention head in one layer, or the average over attention heads and all layers, or what else is it exactly? And for what model?



Eq 3, how is this self-information log prob defined? Is this the attention score A? Why do you write P? Shouldn't this be A?

Sec 3.1 "concatenate them with the input query for parallel encoding" - what exactly does this mean? What is the input query for parallel encoding? Where do I see the input query in the formula?


Sec 3.1 Chunk Eviction: What is this about? "retaining only the most relevant chunks" - what does this mean? What exactly is kept? The KV? But then, how is this different to the KV Cache Eviction?

Sec 3.1 KV Cache Eviction: How is this different now to the previously described chunk eviction?

Sec 3.1 KV Cache Eviction: What exactly is parallel about it? The previous chunk eviction is not parallel or also parallel?

Sec 3.1 KV Cache Eviction: Why is Flash Attention relevant here? Flash attention is just a specific implementation.

Sec 3.1 KV Cache Eviction: "Since it is hard to obtain the complete attention distribution from Flash Attention" - what do you mean? You don't get the attention weights out of flash attention? Why is this relevant? Then just don't use flash attention but something else? Or implement this yourself? Or modify flash attention?

Sec 3.1 Attention Calibration: "we evict tokens with excessively high attention scores": Evict means, you only use those values with high att scores, or you exclude those and only use the others? From eq 7, it looks like you only use those values with high att scores. But why is this helpful? How does this calibrare the attention distribution?

Table 1, 2: What exactly is "Ours" (without the calibration R_h and compression R_l)?

**Relation To Broader Scientific Literature:**

The presented method is relevant for length extrapolation.

**Theoretical Claims:**

I did not check in detail.

---

> ### Author Rebuttal · Authors · 2025-03-31
>
> Thank for your suggestions!
>
> Q1:**For general inquiries such as typos and others.**
>
> - We promise to release our github and will correct typo error in future paper version.
> - Regarding citation issues, the NTK-by-parts position interpolation method was not proposed in a formal paper. It was also cited from a GitHub repository in other papers, with reference [7] in the reference paper [1].
>
> Q2:**Fig 1, is that one selected attention head in one layer, or the average over attention heads and all layers, or what else is it exactly? And for what model? Eq 3, how is this self-information log prob defined? Is this the attention score A? Why do you write P? Sec 3.1 "concatenate them with the input query for parallel encoding" - what exactly does this mean? What is the input query for parallel encoding? Where do I see the input query in the formula?**
> - The attention head we show here is the distribution change of the 21st head of layer 1, and the model is meta-llama/Llama-2-7b-chat-hf.
> - Self-information is not attention score; essentially, it is the negative log-likelihood of the question predicted based on the chunk input to the LLM. It reflects the model's confidence in predicting the query based on the context.
> - We concatenate the input question(X^q in the formula) with each chunk and encode them in parallel. The query serves two purposes: first, to assess the importance of each chunk and decide if its KV should be stored in the GPU’s priority queue to manage memory; second, to evict tokens from the chunk’s KV cache, optimizing memory usage and throughput. In short, the query is primarily used for compression.
>
> Q3:**Sec 3.1 Chunk Eviction: What is this about? "retaining only the most relevant chunks" - what does this mean? how is this different to the KV Cache Eviction? What exactly is parallel about it? The previous chunk eviction is not parallel**
> - Chunk eviction means removing the entire KV cache of a chunk from a priority queue based on self-information scores, while KV Cache Eviction refers to removing tokens from a specific layer and head within a chunk.
> - The token-level KV cache eviction process is parallel across chunks, as it depends on the cumulative attention distribution of the query across layers and heads without dependencies between chunks. In contrast, traditional KV cache eviction methods such as H2O [2] have no extrapolation capability and can only handle KV caches with a fixed context length.
> - The eviction of chunks depends on whether the priority queue has reached its maximum size, usually determined by the GPU memory. When the size is reached, chunks completing self-information calculation and KV cache eviction compare their self-information size with those in the queue to decide retention or eviction. This comparison is typically done sequentially.
>
> Q4:**Why is Flash Attention relevant here? Flash attention is just a specific implementation.**
> - We want to emphasize that flash attention provides faster computation and memory utilization. FlashAttention does not return the complete attention distribution matrix for each layer of the model[3], thus making it unable to perform KV cache eviction. To address this issue, we calculate the inter-chunk attention matrix for each chunk and its corresponding query block at each layer separately, based on formula (5).
>
> Q5:**Since it is hard to obtain the complete attention distribution from Flash Attention" - what do you mean? Sec 3.1 Attention Calibration: "we evict tokens with excessively high attention scores": Evict means, you only use those values with high att scores, or you exclude those and only use the others? What exactly is "Ours" (without the calibration R_h and compression R_l)?**
>
> - According to [3], FlashAttention doesn't return the attention distribution, preventing us from evicting the KV cache based on attention scores. To solve this, we use the last 8 tokens from the query chunk and multiply it with the chunk's value matrix to quickly estimate a cumulative attention distribution for evicting tokens using Pytorch.
> - Equation 7 represents that we evict tokens with abnormally high scores. At the same time, we also evict tokens with low attention scores according to Equation 6. In summary, the eviction strategy removes tokens with either abnormally high or the lowest attention scores.
> - We calculate the full attention distribution for each head using flash-attention. To prevent abnormal patterns, we first approximate the attention distribution with Equation 5, allowing us to evict tokens that would gather excessive attention.
> - 'Ours' represents the case without any KV cache compression or attention calibration, only with the chunk eviction mechanism.
>
> **Reference**:
> [1] YaRN: Efficient Context Window Extension of Large Language Models.
> [2] H2O: Heavy-Hitter Oracle for Efficient Generative Inference of Large Language Models.
> [3] https://github.com/Dao-AILab/flash-attention/issues/1357

---

> > ### Comment · Reviewer_JBTb · 2025-04-02
> >
> > Thank you for the rebuttal.
> >
> > I still think that you should not convolute technical limitations because of certain implementations you use (e.g. FlashAttention) with the actual conceptual methods. But this is mostly a matter of formulation.
> >
> > I also still see it as a very big problem that you do not plan to release your source code.

---

> > > ### Author Response · Authors · 2025-04-04
> > >
> > > Thank you for your response and feedback.
> > >
> > > Q1: **I still think that you should not convolute technical limitations because of certain implementations you use (e.g. FlashAttention) with the actual conceptual methods. But this is mostly a matter of formulation.**
> > >
> > > We agree with your point. Since ICML cannot update the PDF, we will make the required changes in future versions.
> > >
> > > Q2: **I also still see it as a very big problem that you do not plan to release your source code.**
> > >
> > > Do you want us to provide the anonymous repository link directly here? We assure you that if the paper is accepted, we will definitely release our source code.
> > >
> > > If you think that we have addressed all of your concerns, may we kindly ask you to consider increasing the score? Thanks

---

### Decision · Program_Chairs · 2025-05-01

**Decision:**

Accept (poster)

**Comment:**

This paper introduces a training-free approach to generalizing LLMs to longer context lengths. The method computes local attention on chunks of the sequence, and uses this to evict tokens and chunks that have a low score according to their metric. Finally, the remaining chunks undergo global attention.

* Reviewers appreciate that the approach is training-free but still leads to performance improvements on length extrapolation, while improving efficiency and memory usage.
* One of the reviewers encouraged the authors to include comparisons to more recent models, and the authors have included in their response a comparison of their approach applied to the Llama 3 and Qwen models. We encourage the authors to include these in the final submission with appropriate discussion.
* We additionally encourage the authors to address parts the reviewers found unclear, and to include the comparison to STAR and APE. We recognize these were not published before the submission deadline, but discussing them and comparing to them will make this a stronger paper that is easier to contextualize with the literature.

Based on the reviews and author responses, we recommend this paper for acceptance.